# Long-term hepatitis B virus infection of rhesus macaques requires suppression of host immunity

Sreya Biswas[1], Lauren N. Rust[1], Jochen M. Wettengel[1,2], Sofiya Yusova [1], Miranda Fischer[3], Julien N. Carson[1], Josie Johnson[3], Lei Wei [4], Trason Thode [5], Mohan R. Kaadige[5], Sunil Sharma[5], Majd Agbaria[6], Benjamin N. Bimber [3], Thomas Tu[7,8], Ulrike Protzer [2], Alexander Ploss [4], Jeremy V. Smedley [3], Gershon Golomb [6], Jonah B. Sacha[1,3,9] & Benjamin J. Burwitz[1,3,9 ✉]

Hepatitis B virus has infected a third of the world's population, and 296 million people are living with chronic infection. Chronic infection leads to progressive liver disease, including hepatocellular carcinoma and liver failure, and there remains no reliable curative therapy. These gaps in our understanding are due, in large part, to a paucity of animal models of HBV infection. Here, we show that rhesus macaques regularly clear acute HBV infection, similar to adult humans, but can develop long-term infection if immunosuppressed. Similar to patients, we longitudinally detected HBV DNA, HBV surface antigen, and HBV e antigen in the serum of experimentally infected animals. In addition, we discovered hallmarks of HBV infection in the liver, including RNA transcription, HBV core and HBV surface antigen translation, and covalently closed circular DNA biogenesis. This pre-clinical animal model will serve to accelerate emerging HBV curative therapies into the clinic.

[1] Vaccine & Gene Therapy Institute, Oregon Health & Science University, Beaverton, OR 97006, USA. [2] Institute of Virology, Technical University of Munich / Helmholtz Zentrum München, München 81675, Germany. [3] Oregon National Primate Research Center, Oregon Health & Science University, Beaverton, OR 97006, USA. [4] Department of Molecular Biology, Princeton University, Princeton, NJ 08544, USA. [5] Translational Genomics Research Institute, Phoenix, AZ 85004, USA. [6] Institute for Drug Research, School of Pharmacy, Faculty of Medicine, The Hebrew University of Jerusalem, Jerusalem 12272, Israel. [7] Storr Liver Centre, Westmead Clinical School and Westmead Institute for Medical Research, Faculty of Medicine and Health, The University of Sydney, Westmead, NSW 2145, Australia. [8] Centre for Infectious Diseases and Microbiology, Marie Bashir Institute for Infectious Diseases and Biosecurity, University of Sydney at Westmead Hospital, Westmead, NSW 2145, Australia. [9] These authors jointly supervised this work: Jonah B. Sacha, Benjamin J. Burwitz. ✉email: burwitz@ohsu.edu

Hepatitis B virus (HBV) infection is a major cause of morbidity and mortality worldwide, causing nearly 1 million deaths annually[1]. While there is an effective vaccine available, 5–10% of vaccinated individuals are non-responders and there are large geographical regions where vaccination rates remain low[1,2]. The vast majority (~95%) of adults successfully clear acute infection, but infants and young children are particularly susceptible to chronic infection[1]. Those with chronic infection periodically experience liver immune flares that cause progressive damage over the course of decades. Approximately 20–30% of chronically infected individuals will eventually deal with cirrhosis and/or hepatocellular carcinoma[1]. To reduce viral replication and avoid liver immune flares, patients are often treated with nucleoside analogues (NA) inhibiting reverse transcriptase or pegylated-IFNα. However, these treatments are rarely curative and NA must be taken long-term, often for life[3,4]. Importantly, patients on long-term NA still remain at a significantly elevated risk of progressing to hepatocellular carcinoma[5].

In contrast to long-term treatments designed to reduce HBV replication, a major focus of current research is to develop innovative therapies to cure the infection. To eliminate HBV from the liver, treatments must purge covalently closed circular DNA (cccDNA) from hepatocyte nuclei, as cccDNA is the major template for viral gene transcription. HBV therapies currently in the clinical research pipeline are focused on direct viral targeting or modulating the immune system to suppress and clear chronic HBV infection. Direct viral targeting therapies include, but are not limited to NAs, HBV-specific siRNA, CRISPR/Cas9 targeting cccDNA, core allosteric modulators that inhibit nucleocapsid assembly, and the entry inhibitor Bulevirtide (Hepcludex) that is already licensed for the treatment of hepatitis delta virus[6]. In contrast to antivirals, immunotherapies currently under investigation are designed to harness the immune system to better target HBV infected hepatocytes and include immune stimulation with pattern recognition receptor agonists, checkpoint inhibitor blockades, therapeutic vaccines, and adoptive T cell therapy[6]. Unfortunately, while most of these therapies have been tested in murine models, there is no available preclinical non-human primate model of chronic HBV infection to support their further testing and clinical translation.

Rhesus macaques (RMs) have been used for decades as the gold standard for the development of pre-clinical therapeutics to treat degenerative, genetic, age-associated, and infectious human diseases[7–10]. We previously reported that expression of the human HBV receptor, sodium taurocholate co-transporting polypeptide (hNTCP), on RM hepatocytes facilitates HBV infection in vitro and in vivo[11]. However, our previously reported RM HBV model achieved only low levels of viral replication and liver dissemination.

Here, we present a robust second-generation model of extended HBV infection in RMs that is defined by persistent HBV viremia accompanied by hepatitis surface antigen (HBsAg) and e antigen (HBeAg) in the serum for more than six months. We longitudinally biopsied the livers of infected animals and confirmed viral replication by RT-qPCR, immunofluorescence, and in situ RNA hybridization. Thus, we have established the only currently available non-human primate model of robust, long-term HBV infection.

## Results

**RMs exhibit transient viremia following HBV challenge**. We previously demonstrated that vector-mediated expression of hNTCP in the liver of RMs is sufficient to render them susceptible to HBV infection[11]. However, HBV replication was minimal, short-lived, and resolved without the appearance of common serological markers of HBV infection, including HBsAg and HBeAg. These shortcomings significantly reduced the impact and relevance of the model to the clinical setting. In the present study, we aimed to improve the levels of HBV replication and to determine if RMs are susceptible to long-term HBV infection. Given that human infants and young children are highly susceptible to chronic HBV infection, we selected a RM cohort that ranged in age from 5 days to 16 months old to increase our odds of obtaining chronic infection. We injected five juveniles (>11 months old) and three infants (<10 days old) intravenously with replication-incompetent Ad5 (Ad, $N = 5$), helper-dependent Ad5 ($N = 1$), or AAV8 ($N = 2$) expressing hNTCP under the liver-specific transthyretin receptor (TTR) promoter (Fig. 1A, Table S1). Juvenile RMs were placed on a short-term, daily, immunosuppressive tacrolimus regimen (0.04 mg/kg) to drive sustained HBV infection (Fig. 1A). In addition to tacrolimus, we injected all juvenile RMs with liposomal alendronate (1 mg/kg IV) 24 h prior to viral vector administration to deplete Kupffer cells in the liver, a technique we have previously shown as effective in RMs (Fig. 1A)[12]. Kupffer cells have been shown to take up 90% of viral vectors within the first 24 h following high-dose intravenous injection[13]. Therefore, Kupffer cell depletion leads to higher transduction of hepatocytes, a phenomenon which has been shown previously in mice[14]. Finally, high-dose viral vector administration has been shown to induce strong innate immune responses[13]. Therefore, we included dexamethasone injections (1 mg/kg IM) at −12, −4, and +4 h postviral vector administration to dampen the innate immune response, as shown previously in mice[15]. Seven days later, we challenged intravenously with $1 \times 10^9$ genomic copies (gc) HBV genotype D serotype ayw (hereafter simply referred to as HBV)(Fig. 1A). We then measured HBV infection in the blood and liver over the course of ≥10 weeks. HBV serum viral loads (sVL) were detectable in all RMs with peaks ranging from $3.9 \times 10^3$ to $2.6 \times 10^6$ (Fig. 1B), however, sVL were cleared by 8 weeks post-infection (wpi) in all RMs. In addition, we detected serum HBsAg in three of five RMs, although detection was transient similar to sVL (Fig. S1). We next biopsied the livers of juvenile RMs at 6 wpi and at euthanasia (≥10 weeks) to assess HBV core antigen (HBcAg) expression, hNTCP receptor expression, and HBV DNA loads. We discovered HBcAg-positive nuclei at 6 wpi in RM 36651, the animal with the highest peak sVL, but not in any of the other liver biopsies (Fig. S2). We also found hNTCP expression in the livers of all animals at 6 wpi and at the time of euthanasia (Fig. 1C). Importantly, the level of hNTCP expression we detected at both timepoints was greater than or equal to the hNTCP expression levels we have previously shown to support detectable sVL[11], but lower than measured in a HepG2-hNTCP cell line ($1.28 \times 10^7$ copies/100 ng) with the exception of RM 36651 at 6 wpi. Despite this sustained hNTCP expression, we did not detect HBV DNA in the livers of RMs at euthanasia (Fig. 1D).

We also monitored changes in serum alanine transaminase (ALT) levels following HBV challenge, since immune clearance of HBV from the liver is often associated with liver dysfunction[16]. All five juvenile RMs showed elevated levels of serum ALT, concomitant with a drop in HBV viral load to undetectable levels (Fig. 1B, E). This temporal relationship between elevated serum ALT and clearance of circulating HBV DNA supported our nascent hypothesis that transient viremia in RM was due to immune-mediated clearance of HBV infection.

**Transient viremia is associated with adaptive immune responses**. To further define the immune responses occurring in juvenile RMs, we assessed the levels of humoral anti-HBs

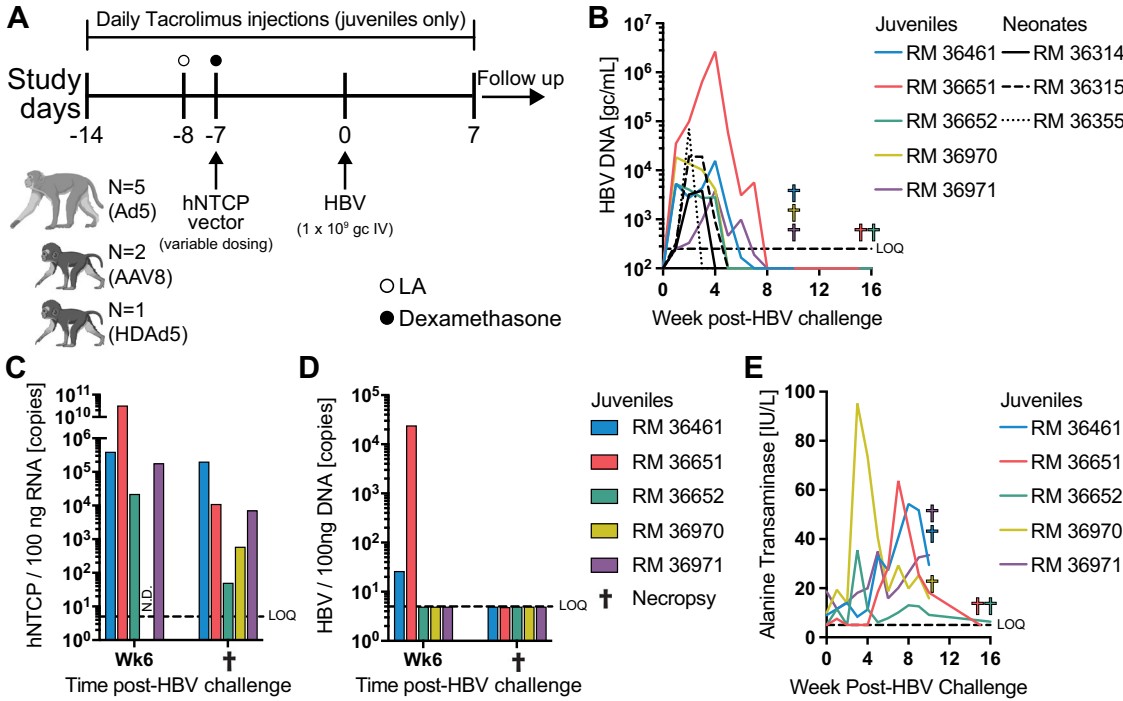

**Fig. 1 Transient HBV infection in RM. A** Timeline for HBV infection of RM. **B** sVL in juvenile and infant RM. **C** Liver hNTCP RNA quantification by RT-qPCR. **D** Liver HBV DNA quantification by qPCR. **E** Serum ALT concentrations. Macaque clip art created with BioRender.com. Source data are provided as a Source Data file.

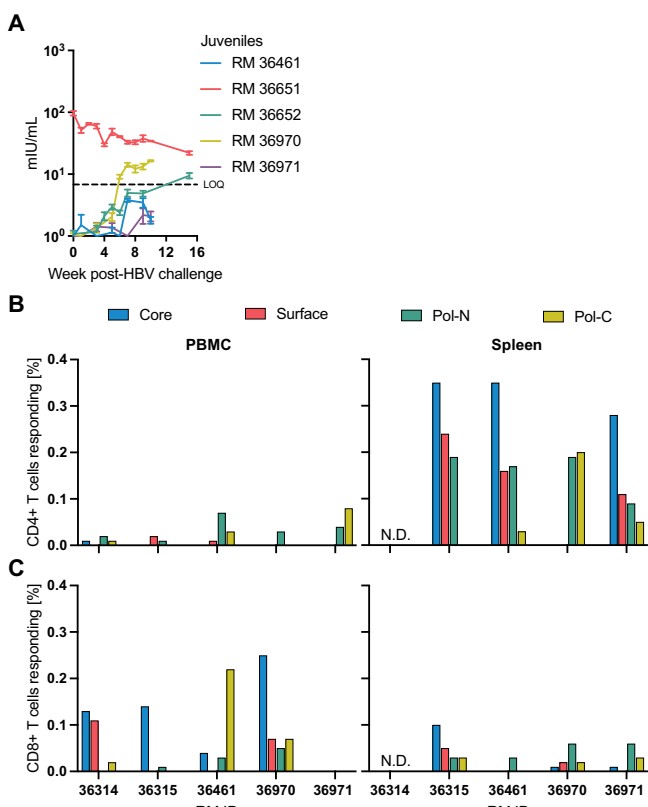

**Fig. 2 HBV-specific adaptive immune responses in RM. A** Anti-HBs concentration in serum. **B** HBV-specific CD4 + and CD8 + T cell responses in blood, mesenteric lymph nodes, and spleens of HBV-infected RM. Data are presented as mean values + / − SD. Source data are provided as a Source Data file.

longitudinally, as well as HBV-specific T cell responses at the time of euthanasia. We detected anti-HBs seroconversion in two of the five RMs (Fig. 2A), although the lack of anti-HBs seroconversion in the remaining three RMs may simply have been due to the relatively short period between HBV clearance and euthanasia. Notably, RM 36651 tested positive for anti-HBs on the day of HBV challenge, although this response was not boosted following HBV infection and did not preclude HBV infection and replication, indicating that this may be a cross-reactive antibody or non-specific binding in the assay. Indeed, although it is well documented that RMs are not susceptible to HBV infection, 6 of 23 HBV-naïve serum samples from the ONPRC colony reacted with the assay (Fig. S3). Thus, for all future studies we screened RMs prior to assignment.

In addition, we measured T cell responses against overlapping peptide pools spanning HBcAg, HBsAg, and HBV polymerase antigen in peripheral blood mononuclear cells (PBMC) and spleen. We detected both CD4 + and CD8 + T cell responses against HBV in multiple animals and tissues (Fig. 2B, C). These results indicated that HBV infection in RMs is cleared through an HBV-specific immune response, since Ad-hNTCP expression is sustained following HBV clearance (Fig. 1C). Therefore, we next studied the duration of HBV replication in immunosuppressed RMs.

**RMs exhibit sustained HBV viremia in the presence of immunosuppression**. We found that despite continued hNTCP expression, HBV infection was cleared in infant and juvenile RMs. We reasoned that this was due to either a species-specific self-limiting infection or immune clearance. Given our results indicating that HBV-specific immunity may be responsible for transient infection in juvenile RMs (Fig. 2), we next asked whether immunosuppression during acute HBV infection leads to sustained viral replication. We subjected three juvenile RMs to a stringent immunosuppression regimen previously shown to avoid graft-versus-host disease in a macaque allogeneic stem cell

transplant model[17]. A detailed description of all administrations and procedures for this cohort can be found in Table S2. This regimen consisted of daily tacrolimus injections (IL-2 inhibitor, 0.06–0.08 mg/kg IM) and intermittent belatacept injections (CD80/CD86 blockade, 20 mg/kg IV) beginning prior to Ad-hNTCP administration and lasting for 126 days post-HBV challenge. We again administered liposomal alendronate to deplete Kupffer cells and dexamethasone to reduce inflammation in the liver prior to Ad-hNTCP injection (Fig. 3A, Table S2, Fig. S4A). We then challenged intravenously with HBV ($1 \times 10^9$ gc) and followed markers of infection in the blood and liver for up to 92 wpi. In support of our hypothesis that an immune response is responsible for HBV clearance in RMs, we measured sustained viremia in all three animals for greater than six months, the clinical definition of chronic HBV infection (Fig. 3B). We observed peak sVL at 6 wpi, all of which were higher than those observed in transiently infected RMs (Fig. 3B, mean range: $3.37 \times 10^6$–$3.48 \times 10^8$). In support of our sVL data, we detected HBsAg and HBeAg in all three RMs (Fig. 3C, D). Notably, sVL, HBsAg, and HBeAg all fell precipitously following tapering and cessation of immunosuppression. These drops were associated with ALT flares in all three RMs (Fig. 3E). Thus, all markers pointed to an immunosuppression-delayed immune response to HBV. Indeed, longitudinal measures of anti-HBc and anti-HBs revealed seroconversion in all three RMs, concomitant with loss of HBsAg (Fig. 3C, F, G). We also observed a trend between the duration of detectable sVL and the magnitude of the anti-HBs response, although this was not statistically significant (Fig. S4B). Notably, sVL were detectable for months in RM 37014 following anti-HBs seroconversion, a status analogous to clinical occult infection.

In order to more thoroughly characterize HBV immunity, we performed longitudinal IFNγ ELISpot assays on PBMC collected from these RMs before and after immunosuppression. We found no responses against HBsAg peptide pools in these ELISpot assays (Fig. 3H–J). Additionally, we followed activation of peripheral T cells before and after the termination of immunosuppression through staining for the nuclear transcription factor Ki67. In contrast to the IFNγ ELISpot data, we measured a large increase in Ki67 expression in both CD4 + and CD8 + T cells following the tapering of immunosuppression (Fig. 3K). Thus, while HBV-specific T cells were not detected in the PBMC, we did discover generalized T cell activation indicative of a mounting immune response.

We next performed a series of experiments to characterize the stringency of anti-HBV immunity in our model, the timing of which is summarized in Table S2. First, we depleted CD8α + cells in RM 36901 at 37 wpi to determine if loss of CD8 + T cells and NK cells would lead to a resurgence of viremia, but found no evidence of HBV recrudescence following depletion (Fig. 3B and Fig. S4C). Second, we boosted RM 37534 with Ad-hNTCP at 52 wpi following confirmation that her serum did not neutralize Ad5 (Fig. S4D). We then rechallenged RM 37534 with HBV at 53 wpi and monitored for reinfection. In support of acquired anti-HBV immunity, RM 37534 exhibited no viremia after rechallenge (Fig. 3B). Finally, we boosted RM 37014 with Ad-hNTCP at 60 wpi, a time point where active HBV replication was observed by sVL (Fig. 3B). We again confirmed no Ad5 neutralizing antibodies prior to boost (Fig. S4D). Despite this re-administration of Ad-hNTCP, we observed no increase in sVL, indicating some level of immune control over viral replication.

**RMs exhibit classical markers of HBV infection in the liver**. These results clearly demonstrate that HBV infection in RMs is cleared through an adaptive immune response, similar to the majority of HBV-infected human adults. However, an effective

RM HBV infection model will also recapitulate the markers of HBV infection seen in the human liver. To assess the breadth and localization of HBV infection in RM livers, we biopsied the livers every 4 weeks for the duration of the study. We first confirmed hNTCP RNA in the livers by RT-qPCR and found extended expression (Fig. 4A). We also detected high levels of HBV DNA and RNA in the livers, and these levels correlated closely with both blood markers of infection and with hNTCP RNA expression in the liver (Fig. 3, Fig. 4B, C, and Fig. S5). Remarkably, RM 37014 exhibited sustained levels of HBV DNA and RNA in the liver following tapering and cessation of immunosuppression, corresponding with the animal's sVL (Fig. 3B and Fig. 4B, C). Finally, to confirm diffuse infection across the liver we collected three liver lobes from RM 37014 and 37534 at euthanasia and measured HBV DNA and RNA. We found similar levels of HBV DNA and RNA across all liver lobes examined (Fig. 4D, E).

Next, we optimized immunofluorescent staining protocols and detected widespread HBV infection in the liver, represented by the hallmark cytoplasmic HBsAg (red) and nuclear HBcAg (pink) staining (Fig. 4F). We also included staining for macrophages (yellow) and confirmed that Kupffer cells showed no signs of HBV replication (Fig. 4F). To further confirm infection, we used an HBV-specific RNAscope panel to identify cells with active HBV transcription and showed that many of these cells also stained positive for nuclear HBcAg (Fig. 4G). Utilizing our longitudinal liver biopsies to quantify the frequency and total counts of cells staining for nuclear HBcAg, we discovered a broad distribution of HBcAg-positive nuclei early in infection during immunosuppression, and these levels tapered following cessation of immunosuppression (Fig. 4H and Fig. S6). Remarkably, HBcAg-positive nuclei were still observed at the time of euthanasia (92 wpi) in RM 37014, consistent with the HBV DNA and RNA levels detected at this time point (Fig. 4B, C and Fig. S6).

Given the generalized T cell activation we observed in the blood post-immunosuppression (Fig. 3K), we next measured infiltration of CD3 + T cells into the liver. Notably, we found increases in CD3 + T cell counts immediately following tapering of immunosuppression (Fig. 4I). However, the increase in CD3 + T cell counts was blunted in RM 37014 (Fig. 4I), These data correspond well to the extended viremia observed in RM 37014 and suggest that clearance of HBV may be T cell-mediated.

Finally, given the focal importance of cccDNA to chronic infection, we performed qPCR to confirm its presence in the livers of HBV-infected RM. We utilized a highly-specific cinq-PCR and found cccDNA in the livers of all three RMs, further validating the infection model (Fig. 4J)[18]. Levels of cccDNA rose and fell with similar kinetics to the sVL in all three animals (Figs. 3B and 4J). These data were further confirmed by traditional T5 exonuclease-based qPCR detection of cccDNA (Fig. S7A), and by showing a lack of cccDNA in matched serum HBV DNA samples (Fig. S7B). Finally, we necropsied an unrelated HBV-infected RM at peak viremia ($2.52 \times 10^7$ gc/mL), collected the liver, and performed a Hirt DNA extraction for cccDNA Southern blot analysis. The Southern blot detected cccDNA that was linearized by EcoRI, resistant to degradation at 85 °C unless linearized by EcoRI, and resistant to ExoI+III digestion (Fig. S7C). Taken together, these data show that the expected clinical markers of HBV infection are found in the RM liver, and that these markers correlate well with the markers of HBV infection observed in the blood.

**HBV evolution in RMs**. HBV is a pararetrovirus with an error-prone reverse transcription step ($1.5 \times 10^{-5}$–$5 \times 10^{-5}$ nucleotide substitutions per site per year) in its replication cycle[19]. This leads

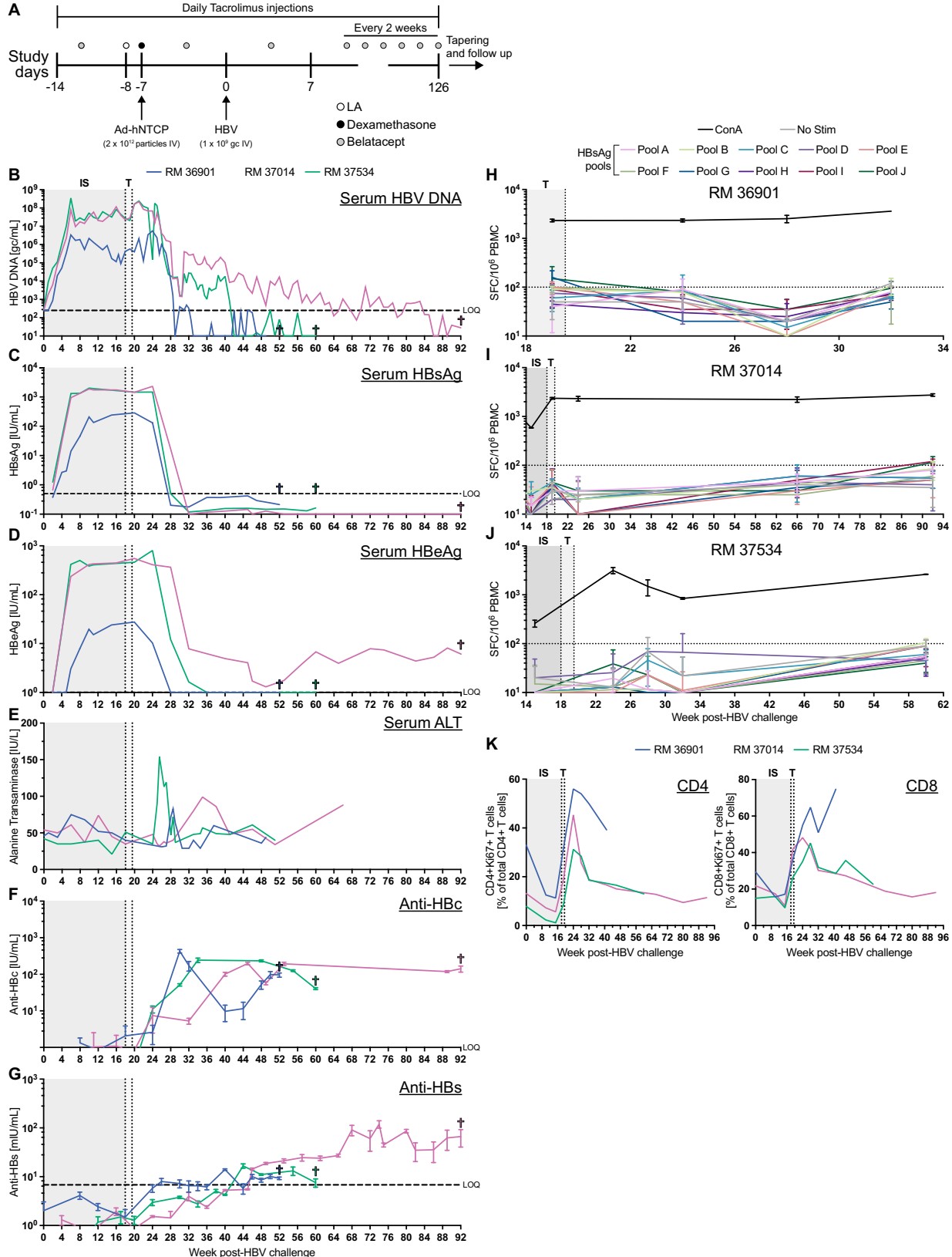

**Fig. 3 Sustained HBV replication in RM on immunosuppression. A** Timeline for HBV infection and immunosuppression of RM. **B** HBV DNA levels in serum during and postremoval of immunosuppression. **C** HBsAg levels in serum. **D** HBeAg levels in serum. **E** ALT levels in serum. **F** Anti-HBc IgG quantification in serum. **G** Anti-HBs IgG quantification in serum. **H–J** Longitudinal measurements of anti-HBsAg T cells by IFNγ ELISpot. **K** Frequency of Ki67-expressing CD4 + and CD8 + T cells in the PBMC. SFC = spot forming cell. Data are presented as mean values + / − SD. Source data are provided as a Source Data file.

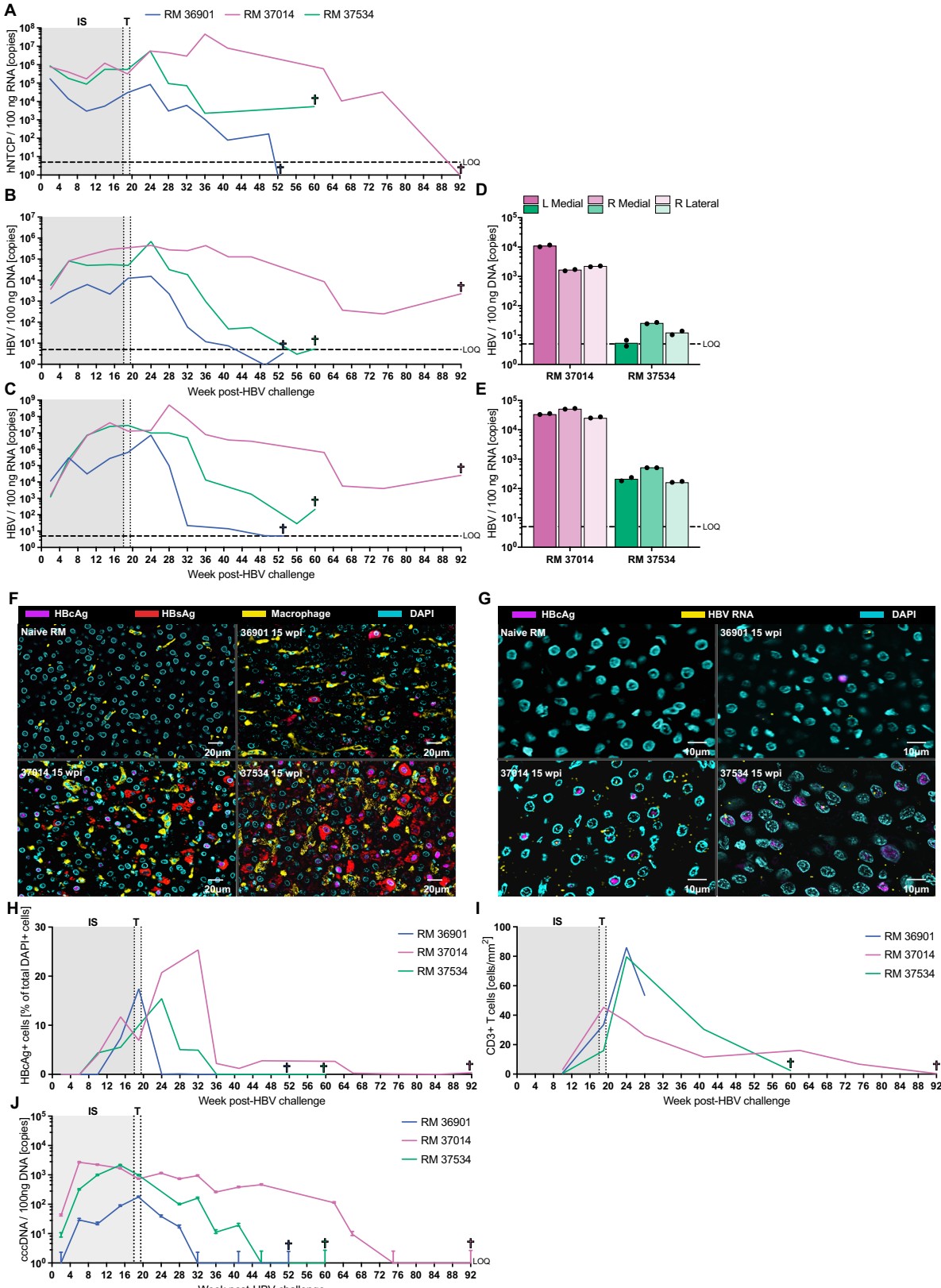

**Fig. 4 Assessment of HBV infection in RM livers. A** hNTCP expression in the liver is consistent over time following HBV infection. **B** HBV DNA levels are consistent over time following HBV infection. **C** HBV expression is consistent over time following HBV infection. **D** Distribution of HBV DNA in the liver lobes at euthanasia. **E** Distribution of HBV RNA in the liver lobes at TOD. **F** Immunofluorescence shows wide-spread HBc (green) and HBs (red) infection of hepatocytes, but not Kupffer cells (yellow). **G** Dual fluorescent RNAScope-IHC staining shows HBV RNA (red) and HBcAg expression (green) in the liver of infected RM. **H** Frequency of HBcAg-positive nuclei in the liver of HBV-infected RM. **I** Quantification of CD3 + T cells in the liver of HBV-infected RM. **J** cccDNA levels in RM liver following HBV infection. Data are presented as mean values +/− SD. Three independent experiments were conducted for panels F-G with similar results. Source data are provided as a Source Data file.

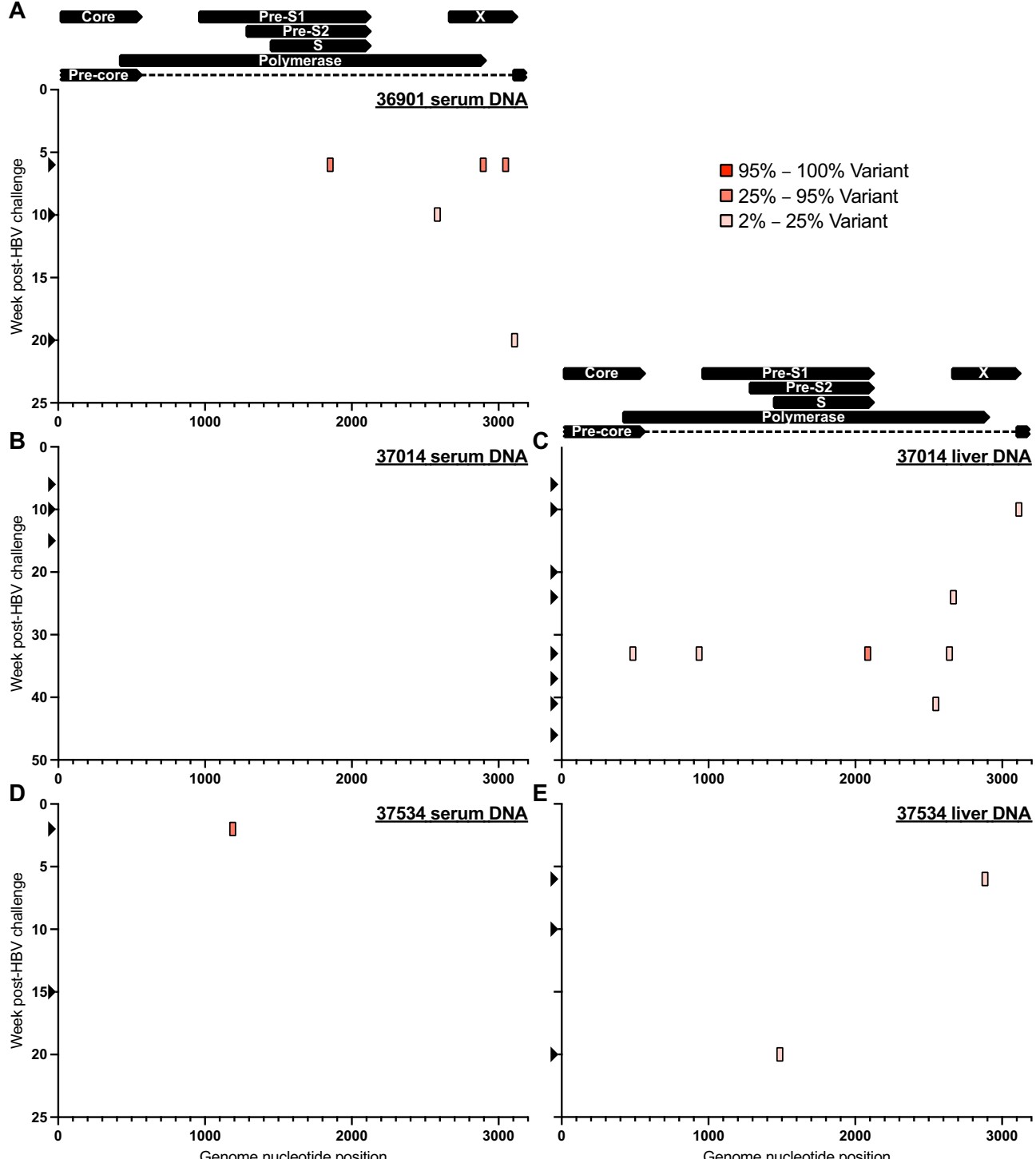

**Fig. 5 HBV sequencing of serum and liver DNA from HBV-infected RM.** Shows genetic variation (single nucleotide polymorphisms) across the HBV genome (position 1 is start codon of HBV core ORF). Black arrows on y-axes indicate time points sequenced. Source data are provided as a Source Data file.

to HBV genetic variation both within and across infected individuals, and this evolution can directly impact clinical care and outcomes[20]. We next defined the HBV genetic variation seen in the serum and liver biopsies of HBV-infected RMs. We utilized a previously published unbiased approach to sequence HBV from these tissues, which includes rolling circle amplification of the HBV genome, tagmentation, and Illumina sequencing[21]. Strikingly, we found little to no genetic variation in the serum or liver samples from the three infected RMs across multiple time points,

including during immunosuppression and peak viral replication (black arrows on y-axes, Fig. 5). RM 36901 harbored three intermediate frequency variants (25–95% of reads) at 6 wpi in serum-derived HBV (Fig. 5A). The variant at position 1855 (40%) was synonymous in the Pol open reading frame (ORF), but nonsynonymous in the Pre-S1/S2/S ORF (T80I). In addition, the variant at position 2895 (40.4%) was synonymous in the X ORF, but nonsynonymous in the Pol ORF (K810R). The final variant at position 3048 (41.3%) was synonymous in the X ORF.

Interrogation of later time points revealed reversion of all three variants, with only two transient low-frequency mutations found across subsequent time points (Fig. 5A). In contrast to RM 36901, serum-derived HBV from RM 37014 was completely wild-type (no mutations above 2% frequency) at all time points examined (Fig. 5B). However, examination of HBV sequences obtained from the liver biopsies of RM 37014 showed greater diversity, although all but one variant were low frequency (Fig. 5C). The sole intermediate frequency variant at position 2,084 (26.6%) was synonymous in the Pre-S1/S2/S ORF, but nonsynonymous in the Pol ORF (T560A). However, this mutation was found at a single time point and did not persist (Fig. 5C). Finally, RM 37534 showed a similar lack of HBV diversity in the serum and liver, with a single intermediate frequency variant at position 1189 (50.5%) that was synonymous in the Pol ORF, but non-synonymous in the Pre-S1 ORF (P81Q, Fig. 5D, E). Taken together, these results show that despite sustained HBV replication for the first 6 months of infection, there was little to no variation within the HBV quasispecies.

## Discussion

HBV research is currently hamstrung by the lack of an available pre-clinical, non-human primate model. RMs are commonly used for immunological studies, and given their anatomical and physiological similarities with humans, we set out to improve upon our previously reported HBV infection model[11]. Here, we advance the model by showing that administration of immunosuppressants during acute infection can drive sustained HBV replication.

The model faithfully recapitulates the markers of HBV infection in humans, both in the blood and liver. Improvements on our earlier RM HBV model include: 1) detection of circulating HBsAg and HBeAg, 2) molecular and microscopic detection of active viral transcription and translation across the liver, 3) identification of T cell responses to HBV in RMs clearing acute infection, and 4) seroconversion to both HBcAg and HBsAg. However, given the small number of animals utilized and the variability in their outcomes, important questions remain and answers to these questions will govern the future utility of the model.

First, a more comprehensive analysis of HBV immune responses in the liver and blood is needed to further define the model. We show here that both cellular and humoral immune responses are elicited in HBV-infected RMs, but extending these analyses to specifically characterize HBV-specific immunity in the liver will continue to inform improvements to the model. Chronic infection in humans is characterized by weak, narrowly-focused T cell responses and eventual deletion of these T cells over time[22]. In addition, HBV is a stealth virus that minimizes innate immune responses in the liver through downregulation and/or degradation of RIG-I, SMC5/6, and other pattern recognition receptors[23–25]. With these characteristics in mind, future HBV-infected RMs should be monitored for HBV-specific T cell responses long-itudinally and for innate immune activation during and after immunosuppression in the liver. Indeed, the treatment of RMs with Ad-hNTCP prior to HBV infection is a confounding variable, particularly given the immunogenicity of adenoviral vectors in the liver[26]. However, we found that less immunogenic helper-dependent Ad-hNTCP and AAV8-hNTCP also led to transient viremia. Additional avenues should be explored to remove the requirement for viral-vectored hNTCP transduction, either through use of nonviral-vectored delivery or by creation of transgenic RMs that naturally express hNTCP from their germline.

Second, although immunosuppression administration can drive long-term HBV infection, it is also counter to the final goal of providing a fully immunocompetent model of infection. Indeed, the immunosuppression employed in this study is highly stringent and has been shown to completely allay graft-versus-host disease and graft rejection in allogeneic stem cell transplant recipients by suppressing T cell activation[17,27]. Therefore, future studies should identify specific immune subsets responsible for transient viremia in RMs. These subsets can then be targeted for transient depletion or HBV immunotolerance as a substitute for generalized immunosuppression. Overall, the largest remaining barrier for the model is the immune clearance of HBV infection in the absence of immunosuppression.

Third, understanding why little to no HBV evolution was observed in RMs will be important as the model is adopted for future testing of immunomodulatory therapeutics. Although immunosuppression during the first 20 wpi could partially explain the lack of viral evolution, due to the lack of immune pressure, it is intriguing that no species-specific adaptations were observed during sustained viral replication. This further supports our and others' findings that viral entry is the single barrier to HBV replication in macaques[11,28]. Unfortunately, the lack of diversity in the HBV quasispecies also indicates that in vivo adaptation of HBV PreS1 to allow for entry into RM hepatocytes using the endogenous macaque NTCP may not be feasible.

Finally, although decreases in sVL and HBeAg are expected during the immune clearance phase of HBV infection, we also detected reductions in HBsAg to below the limit-of-quantitation, concomitant with the emergence of anti-HBs seroconversion. However, sVL and HBeAg remained steady in the serum of RM 37014. This clinical outcome is analogous to occult HBV infection, defined as the presence of liver and/or serum HBV DNA in the absence of serum HBsAg. Nearly 35% of patients with occult infection are positive for anti-HBs, despite ongoing viral replication[29]. The epidemiology of occult HBV infection is complex, since liver biopsies are rarely available and HBV DNA levels in the serum are often near or below the limit of detection. This epidemiological uncertainty is exemplified by review of cross-sectional studies, which predict the prevalence of occult infection to be between 1 and 87% depending on the region of the world and the tests employed[30]. Thus, our RM model may provide new insights into occult HBV infection in an experimentally controlled setting.

Overall, we believe this RM model is an important experimental system, particularly given the absence of chimpanzees and the need for a tractable non-human primate model of HBV infection. While key questions remain regarding pathology and immunology, we show conclusively that HBV-infected RM exhibit all hallmarks of human HBV infection in the blood and liver. Further studies are needed to explore the potential of the model for testing antivirals, and future work will continue to characterize the immune mechanisms of HBV persistence in RM and define the parameters necessary for chronic HBV infection.

## Methods

**Experimental design.** Given the exploratory nature and the binary readout (long-term infection vs. transient infection) of this study, we utilized small animal groups. Randomization was not used to assign animals to their respective experimental groups, and authors were not blinded to assignments. A total of 11 Indian-origin RM were assigned to a neonatal group (36314-F-6 days old, 36315-M-6 days old, 36355-F-5 days old), a short-term immunosuppression group (36461-M-488 days old, 36651-F-404 days old, 36652-M-488 days old, 36970-M-403 days old, 36971-M-371 days old), or a long-term immunosuppression group (36901-F-388 days old, 37014-F-391 days old, and 37534 -F-187 days old). Details of procedures and timing of administrations can be found in Tables S1 and S2. All groups were challenged intravenously with $1 \times 10^9$ gc of HBV (Genbank accession #MN645906). Tacrolimus (Astellas Pharma Inc.) was administered intramuscularly daily at a dose of 0.04 mg/kg (in the short-term immunosuppressed group) and 0.08 mg/kg (in the long-term immunosuppressed group) starting at -14 dpi. Tacrolimus trough levels were measured every 24 h and confirmed to be between 5 and 15 ng/mL in the blood. In the short-term immunosuppression regimen, tacrolimus levels were tapered by 0.01 mg/kg starting at 6 dpi and tapering was continued through 8 dpi. As part of the long-term immunosuppression regimen,

tacrolimus levels were tapered by 20% starting on 126 dpi and tapering was continued through 138 dpi. Liposomal alendronate (1 mg/kg) was administered intravenously on -8 dpi. Dexamethasone (1 mg/kg, Fresenius Kabi USA, LLC, Cat# 500601) was administered intramuscularly at 2 and 12 h prior to intravenous Ad-hNTCP ($3.5 \times 10^{11}$ gc/kg) administration on −7 dpi. Belatacept (20 mg/kg, Bristol-Myers Squibb) was administered intravenously every 7–14 days in RM receiving long-term immunosuppression starting at -10 dpi and continuing through 123 dpi. 36901 was also administered depleting anti-CD8α (Clone MT807R1, Nonhuman Primate Reagent Resource, Cat# AB_2716320) subcutaneously at 5 mg/kg on 260 dpi, followed by three additional 5 mg/kg intravenous doses on 263, 267, and 268 dpi. Animals were cared for at the Oregon National Primate Research Center (ONPRC) with the approval of the Oregon Health & Science University's Institutional Animal Care and Use Committee using the standards of the NIH Guide for the Care and Use of Laboratory Animals. Ex vivo experiments were performed in duplicate where possible. When duplicate samples were not attainable, additional experiments were performed to ensure repeatable results. No outliers were excluded.

**Creation of Ad-hNTCP.** First, the expression cassette TTR-hNTCP-WPRE-bGHpA was synthesized with 5' SpeI and 3' CsiI restriction sites (hNTCP: Genbank accession no. NM_003049.4). Next, Piggybac plasmid PB511 (System Biosciences, Palo Alto, CA) was digested with fast-digest SpeI and CsiI (ThermoFisher, Cat# FD2114). Our synthesized expression cassette was then ligated into the PB511 backbone using T4 DNA ligase (Invitrogen, Cat# 15224017), forming plasmid PB511-hNTCP. To generate the adenoviral entry plasmid, pENTR1A (Invitrogen, Cat# A10462) was digested with SalI (ThermoFisher, Cat# ER0641), end-blunted using the Quick Blunting Kit (New England Biolabs, Ipswich, MA, Cat# E1201L), and dephosphorylated. In parallel, PB511-hNTCP was digested with fast-digest PteI (ThermoFisher, Cat# FD2134) and end-blunted. The PB511-hNTCP insert (containing the hNTCP expression cassette within a Piggybac transposon) was then blunt ligated into pENTR1A using T4 DNA ligase, forming plasmid ENTR-PB511-hNTCP.

Adenoviral vector was generated using the pAD/CMV/V5-DEST Gateway vector system (Thermofisher, Cat# V49320), based on the E1 and E3-deleted human adenovirus, serotype 5. Adenoviral crude stock was generated by transfecting the SmiI-digested pAD-PB511-hNTCP plasmid into 293 A cells (Life Technologies, Cat# R70507) using Lipofectamine 3000 (Thermofisher, Cat# L3000015). Adenoviral crude stock (Ad-hNTCP) was amplified across three passages on HEK293A cells. For in-vivo injection stocks, adenoviral stocks were generated by expanding the virus on 293A cells. The virions were harvested from the cells by taking the cells through three freeze/thaw cycles. The resulting lysates were treated with benzonase nuclease to digest genomic DNA. The virions were purified by ultracentrifugation over two cesium chloride gradients, the first a step gradient[31], the second an isopycnic gradient[32]. The virions harvested from the second gradient were dialyzed against a physiological buffer, 10 mM tris pH 8.0 (Fisher, Cat# BP2473-100), 2 mM magnesium chloride (Sigma Aldrich, Catalog# 7786-30-3), 4% sucrose (w/v) (EMD, Catalog# SX1075-1)[31]. The particle titer of the virus stock was determined by spectrophotometry at absorbance 260 nm. The infectious titer was determined by TCID50 on 293 A cells. The stock was checked for the absence of replication competent virus by a plaque assay/agarose overlay method on A549 cells (Thermo Fisher, Cat# K1679B)[33]. The A549 cells were stained with Thiazolyl Blue Tetrazolium Bromide (Millipore-Sigma, Cat# M2128) to assist visualizing plaques.

**Generation of HBV stocks.** High-titer HBV was produced as we have previously described[11]. Briefly, HBV-containing supernatant from HepAD38 cells was collected every 3–4 days. Purification of HBV was performed via heparin affinity chromatography followed by sucrose gradient ultracentrifugation. HBV stocks were titered by HBV DNA qPCR (see below) for in vivo use.

**HBV DNA and RNA quantification.** Serum viral DNA was extracted from 200 μl using the QiaAmp Minelute Virus Spin kit (Qiagen, Cat# 57704) with a modified version of the manufacturer's instructions. Briefly, protease and lysis buffer were added and the sample incubated at 56 °C for 90 min (instead of the suggested 15 min) while shaking at 750 rpm. All subsequent steps followed the manufacturer's instructions. Total intracellular DNA and RNA were extracted from the liver tissues. Briefly, samples were snap-frozen in lysing matrix tubes (MP Bio, Cat# 116913050-CF) soaked in 1 mL Tri-reagent (Molecular Research Center, Catalog# RN190) and homogenized at 4000 rpm for 30 sec in a Bead Bug™ microtube homogenizer (Millipore Sigma, Catalog# Z763713). First, RNA extraction was performed by addition of 1/10th volume bromochloropropane (Sigma Aldrich, Cat# B9673) to the tri-reagent and then vortexed and incubated at room temperature for 5 min and then spun at $12,000 \times g$ for 5 min to achieve phase separation. 12 μL glycogen (Thermo Fisher, Cat# 10814010) was added to the tubes and the RNA-containing upper aqueous phase was transferred to a fresh tube and placed on ice. Samples for DNA extraction were processed by addition of DNA Backextraction buffer (9.09 g Tris base (Fisher Scientific, Cat# 77-86-1) was added to a 3.75 mL 1 M sodium citrate (Fisher Scientific, Cat# BP327-500) solution and 50 mL 6 M guanidine thiocyanate (Thermo Fisher, Cat# AM9422), then diluted to

a final volume of 75 mL with distilled water) to the sample tubes containing the remaining interphase/organic phase mix after RNA extraction. Samples were vortexed and spun at $12,000 \times g$ for 5 min to achieve phase separation followed by addition of 12 μL glycogen (Thermo Fisher, Cat# 10814010). The DNA-containing upper aqueous phase was transferred to a fresh tube and placed on ice. Next, both the DNA and RNA samples were treated with isopropanol (Thermo Fisher, Catalog# 383920025) mixed gently by inversion and spun at $15,000 \times g$ for 5 min at room temperature. The isopropanol was carefully aspirated and 75% ethanol was added to the pellet. The tube was again spun at $15,000 \times g$ for 5 min and the ethanol wash repeated a second time. All residual ethanol was removed with a micropipette. The pellets were dried at 37 °C on a heat block followed by resuspending the pellets in 100 μL water (RNA) or TE buffer (DNA) (Thermo Fisher, Cat# 12090015). Tubes were placed back in the 37 °C heat block and were shaken at 500 rpm for 15 min. Tubes were gently vortexed and placed on ice for use in assays. Quantity and integrity of the extracted DNA and RNA were assessed on a NanoDrop 2000 Spectrophotometer (Nanodrop Technologies, Catalog# ND2000).

Total HBV RNA was treated with DNase I (Thermo Fisher, Catalog# EN0521) and converted to cDNA using the SuperScript VILO cDNA Synthesis Kit (Thermo Fisher, Catalog# 11754050) according to manufacturer's instructions. Total HBV DNA and cDNA were quantified using TaqMan Fast Advanced Master Mix (Applied Biosystems, Catalog# 4444556). Total HBV DNA/RNA primers and probe were: HBV_qPCR-F (5'-GGCCATCAGCGCCGTGC-3'), HBV_qPCR-R (5'-TGCTGCGAGCAAAACA-3'), and HBV_qPCR-Probe (5'-6FAM-CTCTG CCGATCCATACTGCGGAACTC-TAMRA-3') using an annealing temperature of 62 °C. All thermocycling parameters followed exactly to suggested manufacturer's instructions. All thermocycling and quantification analyses were conducted on an QuantStudio 3 (Applied Biosystems, Catalog# A28567). Quantification was assessed relative to an absolute standard curve using the plasmid pCEP4 with the targeted insert as template.

**hNTCP RNA quantification.** Total intracellular RNA was extracted from the liver tissues as described above. The SuperScript VILO cDNA Synthesis Kit (Thermo Fischer, Catalog# 11754050) was used for cDNA synthesis according to manufacturer's instructions. Before cDNA synthesis samples were treated with DNase I (Thermo Fisher, Catalog# EN0521) to limit rcDNA contamination. Primers used were: hNTCP_qPCR-F (5'-TGCCTCAATGTTCTTCAGCC-3'), hNTCP_qPCR-R (5'-TGTTCATGTTGTTCTTCATC-3') using an annealing temperature of 62 °C. All thermocycling and quantification analyses were conducted on a QuantStudio 3 (Applied Biosystems, Catalog# A28567) using PerfeCTa SYBR Green FastMix, ROX (Quantabio, Catalog# 95055-500) and a 3-step PCR cycling protocol. RNA level quantification was assessed relative to an absolute standard curve using plasmid DNA containing the entire hNTCP sequence. Glyceraldehyde-3-phosphate dehydrogenase (GAPDH) was run as controls to confirm RNA quantity and quality between samples.

**T cell analysis.** HBV-specific CD8 + and CD4 + T cell responses were measured in mononuclear cell preparations from PBMC or spleen from HBV convalescent RM by flow cytometric intracellular cytokine staining, as previously described[34]. Briefly, mononuclear cells were stimulated with pools of overlapping 15-mer peptides corresponding to HBV open reading frames (Genscript; 70% purity) and the costimulatory molecules anti-CD28 clone CD28.2 (BD #556620) and anti-CD49d clone 9F10 (BD #555502) for 1 h, followed by addition of brefeldin A (Sigma-Aldrich) for an additional 8 h. DMSO (vehicle for peptide dilution) stimulation served as a background control. Cells were surface stained for anti-CD4 clone OKT4 (Biolegend #317414) and anti-CD8 clone SK1 (Biolegend #344714), and intracellularly stained with anti-CD3 clone SP34-2 (BD #558124), anti-IFNγ clone B27 (BD #554702), and anti-TNF clone MAb11 (BD #557068). For T cell phenotyping, PBMC was surface stained for anti-CD3 clone SP34-2 (BD #558124), anti-CD4 clone OKT4 (Biolegend #317414), anti-CD8 clone SK1 (Biolegend #344714), anti-CD20 clone 2H7 (Biolegend #302314), and anti-CD14 clone M5E2 (Biolegend #301838) and then stained intracellularly with anti-Ki67 clone B56 (BD #561284) using the FoxP3 staining kit (eBiosciences). Flow cytometric analysis was performed on an LSR-II instrument running FACSDiva software v6.0 (BD Biosciences). Analysis was performed using FlowJo software v10.8.1 (Tree Star).

**IFNγ ELISpot.** EDTA-anticoagulated blood was layered over ficoll, spun at $1800 \times g$ for 30 min, and PBMC isolated from the interface. PBMC from each time point was cryopreserved in Recovery Cell Culture Freezing Media (Thermo Fisher, Catalog# 12-648-010) for subsequent analysis. PBMC were thawed and washed twice in complete media. Pre-coated IFN-γ ELISpot$^{PLUS}$ plates (Mabtech Inc., Catalog# 3421M-4APT-10) were used to detect T cell responses and experiments were conducted per manufacturer's recommendations in triplicate wells as previously described[11]. Each well contained $1 \times 10^5$ PBMC in 100 μl complete media. Cells were incubated with 5 μM 15-mer peptide pools spanning HBsAg (70% peptide purity or greater). Wells were imaged and analyzed on an AID Classic ELISpot reader (Autoimmun Diagnostika GMBH) using an automated algorithm with set parameters for spot size, intensity, and gradient. Based on no stimulation controls, responses less than 100 SFCs per million cells were considered negative (below the limit of detection).

**Ad5 neutralization assay**. HEK293 cells were plated onto a 96 well plate at 80% confluency in complete media. These cells were transduced (MOI = 3) with Ad5 expressing luciferase (Ad5-NanoLuc) following co-incubation of Ad5-NanoLuc with a 1:10 dilution of RM serum samples for one hour at 37 °C. Luciferase activity was determined 72 h post-transduction by transferring 10 μl of supernatant into a white 96-well plate and measuring luminescence on an Infinite 200 plate reader (Tecan) after adding 100 μL of PBS-T (0.1% Tween-20) (Sigma Aldrich, Catalog# P1379) containing a 1:1000 dilution of 1 mM Coelenterazine H (dissolved in acidified methanol).

**cccDNA detection and quantification**. Copies of cccDNA were quantified by cinqPCR (which is compatible with the Genotype D HBV strain used in this project), as previously described with minor modifications[18,35,36]. Between 1 and 3 μg of total liver DNA was inverted using serial HhaI-mediated digestion, T4 DNA ligase-mediated ligation, and XbaI-mediated ligation, exactly as previously described. Given the high HBV DNA replication expected from sVL results, inverted DNA samples were diluted 1:10 in water to ensure samples remained in the linear range of quantification.

The cccDNA inversion reaction produces an inverted 999 bp fragment (nt1801-nt2800) that is amplifiable by the cccDNA primer set, while other HBV DNA forms are not compatible with amplification. The forward, reverse and probe sequences for the cccDNA primer set were: 5'-CACTCTATGGAAGGCGGGTA-3', 5'-ATAAGGGTCGATGTCCATGC-3', and 5'-FAM-AACACATAGCGCACCA GCA-BHQ1-3'. cccDNA copies were normalised to single-copy gene RNase P (quantified by TaqMan™ Copy Number Reference Assay, 4403328, Applied Biosystems). Digital droplet PCR was carried out on 5 μL of diluted inverted DNA as identically as previously described except using ddPCR Multiplex Supermix (BioRad, #12005910) as the buffer[18,35,36]. Droplet data was analysed using QuantaSoft Analysis Pro software v1.4 (Biorad).

In samples where cccDNA was at undetectable levels, the digital droplet PCR was repeated on undiluted inverted DNA. The theoretical lower limit of detection in these cases was 1 cccDNA copy per 150 ng DNA (~25,000 cells).

For traditional qPCR detection of cccDNA, total liver DNA samples and matching serum DNA samples were either digested with T5 Exo (reaction: 5 μL eluted DNA, 2 μL 10X reaction buffer, 1 μL T5 Exo, and 12 μL water) at 37 °C for 1 h and 70 °C for 20 min or left undigested. cccDNA was quantified using the TaqMan Fast Advanced Master Mix (Applied Biosystems, Catalog# 4444556) and all thermocycling and quantification measurements were conducted on a QuantStudio 3 (Applied Biosystems, Catalog# A28567). An optimized two-step qPCR program was established: 95 °C for 15 min followed by 50 cycles of 95 °C for 5 sec (denaturation) and 62 °C for 70 sec (annealing and extension). A plasmid containing the target sequence was used as a standard curve as described above. The following primers and probe were used. cccDNA_qPCR-F (GTCTGTGC CTTCTCATCTGC), cccDNA_qPCR-R (AGTAACTCCACAGTAGCTCCAA ATT) and cccDNA_probe (5'-6FAM-TTCAAGCCTCCAAGCTGTGCCTTGGG TGGC-TAMRA-3').

For cccDNA Southern blotting, 1 gram of snap-frozen liver tissue was thawed and cut into small pieces. The solution containing 10 mM Tris–HCl (Fisher Scientific, Catalog# 1185-53-1), pH 7.5 and 10 mM EDTA (Milipore Sigma, Catalog# 60-00-4) was added to the liver tissue, which was then homogenized by a Dounce homogenizer. The lysate was then centrifuged at 2,500 rpm for 15 min and the pellet was collected and resuspended in 1 mL lysis buffer (10 mM Tris-HCl pH 8.0, 0.625% SDS, and 10 mM EDTA). The lysate was kept at room temperature for 30 min, and 5 M NaCl (Fisher Scientific, Catalog# S271-500) was added to the lysate to reach a final concentration of 1 M. The lysate was incubated at 4 °C overnight and subsequently centrifuged at 12,000 × g for 30 min at 4 °C. The supernatant was collected and extracted by phenol (pH 8.0) twice and phenol-chloroform (pH 8.0, 1:1) once. The supernatant was then supplemented with 0.8 volume of isopropanol and kept at −20 °C overnight. The solution was then spun at 12,000 × g at 4 °C for 30 min, the pellet was washed by 70% ethanol, dried for 5 min, and dissolved in 70 μL nuclease-free water. An aliquot of purified HBV cccDNA was treated with 5 U of ExoI (NEB, Catalog# M0293S), and 25 U ExoIII (NEB, Catalog# M0206S) in 1x Cutsmart (NEB, Catalog# B6004S) buffer at 37°C for 2 hrs.

cccDNA detection by Southern blotting was carried out as previously described[37,38]. The cccDNA samples were resolved on a 1% (wt/vol) agarose gel. The gel was treated with 0.1 M HCl for 10 min, subsequently denatured by 0.5 M NaOH Fisher Scientific, Catalog# AA4578022) and 1.5 M NaCl (Fisher Scientific, Catalog# S271-500), and then neutralized with neutralization buffer (1 M Tris-HCl, pH 7.4 and 1.5 M NaCl). The DNA was then transferred onto a positively charged nylon membrane (Sigma Aldrich) using standard 20X SSC buffer. The DNA was then cross-linked to the membrane by ultraviolet irradiation at 120,000 μJ cm$^{-2}$ (UV stratalinker 1800). The membrane was incubated with the pre-hybridization solution from the digoxigenin (DIG) high prime DNA labelling and detection starter kit II (Sigma Aldrich, Catalog# 11585614910) for 1 h at 42 °C, before hybridized with DIG-labeled HBV probes (558 bp; nucleotides 2625–3182, GenBank accession #U95551.1) at 42 °C overnight. The hybridized membrane was washed three times with washing buffer (0.1 M maleic acid (pH 7.5), 150 mM NaCl and 0.3% (vol/vol) Tween 20), and was subsequently blocked at room temperature for 15 min, and then incubated with alkaline phosphatase-conjugated DIG antibody at a 1:10,000 dilution at room temperature for an hour. The membrane

was washed with washing buffer for 15 min, and was then subjected to chemiluminescent detection using CSPD solution, and the signal was captured on autoradiography film (Thermo Fisher Scientific). The film was scanned with an Epson scanner at a resolution of 400 dpi.

**HBsAg and HBeAg detection by chemiluminescent immunoassay (CLIA)**. HBsAg and HBeAg serum concentrations were determined using the Hepatitis B Surface Antigen (Ig Biotech, Catalog# CL18003) and Hepatitis B Virus E Antigen CLIA kits (Ig Biotech, Catalog# CL18005) according to the manufacturer's instructions. Briefly, 96-well plates pre-coated with either the anti-HBs or anti-HBe monoclonal antibodies were incubated with animal serum, standards, or a negative control. An HRP-conjugated antibody was added to the wells and the samples were incubated for 60 min at 37 °C. Wells were washed and incubated with the kit-provided substrate solution for 10 min at room temperature in the dark. Luminescence was recorded using the SpectraMax M5 spectrophotometer (Molecular Devices, Catalog# 13352). Tests were performed in duplicate and analyzed relative to a standard curve.

**Anti-HBs and anti-HBc ELISA**. Antibodies against HBsAg and HBcAg present in RM serum were quantified using in-house ELISA assays. High-binding, clear, flat-bottom 96-well half-area microplates (Corning, Catalog# 3690) were coated with recombinant HBV surface (Abcam, Catalog# ab91276) or core antigen (Abcam, Catalog# ab115992) at 1 mg/mL in carb/bicarb buffer (ThermoFisher Scientific, Catalog# 28382) at 50 μL per well. Plates were incubated at 4 °C for at least 18 h prior to use. Coated plates were washed with wash buffer (0.05% Tween-20 in 1x PBS) two times then blocked with 150 μL blocking solution per well (10% goat serum in 1x PBS) for one hour at 25 °C. Serum samples were heat-inactivated at 56 °C for 30 min, centrifuged at 10,000 × g for 5 min, and diluted in prepared sample diluent (0.1% skim milk powder (Fisher Scientific, Catalog# NC9121673), 10% BSA (Milipore Sigma, Catalog# 9048-46-8), and 0.05% Tween-20 in 1x PBS). After the plate blocking step, diluted samples were added to the plate at 50 μL per well in triplicate and incubated at 25 °C for 20 min. After incubation the wells were washed five times with 150 μL wash buffer, blotting between washes. Peroxidase-conjugated goat anti-human IgG Fcγ-specific polyclonal antibodies (Jackson ImmunoResearch Laboratories Catalog# 109-035-098) was diluted in prepared sample diluent at 1:5,000 and distributed at 50 μL per well and incubated at 25 °C for 20 min. After incubation with the secondary antibody, the wells were washed five times with 150 μL wash buffer, blotting between washings. Wells were developed with 3,3',5,5'-Tetramethylbenzidine (Southern Biotech, Catalog# 0410-01) at 50 μL per well and incubated at 25 °C for five minutes. Development was halted with 2 M HCl (Sigma Aldrich, Catalog# 13-1683) at 50 μL per well. Plate was read on a VERSAmax Microplate Reader at 450/650 nm (Molecular Devices). The OD450-OD650 values from the anti-HBs ELISA were converted to mIU/mL by comparison against a standard curve generated using human Hepatitis B immune globulin (HBIG) or the WHO international anti-HBc standard (NBISC) for quantification.

**Immunofluorescence for detection of HBcAg and HBsAg**. Liver samples were fixed in 4% paraformaldehyde, processed for paraffin embedding, and cut at 7 μm thickness. The slides were de-waxed in xylene, rehydrated in graded ethanol and then subjected to antigen retrieval in citraconic anhydride (CA) solution (Sigma) (0.1 M, pH 6.0) for 15 min at 110 °C and 5 psi. Slides were cooled, blocked in 0.25% Casein (Sigma, Cat# C7078) solution (in PBS) for 10 min at room temperature and incubated with an anti-HBc rabbit polyclonal antibody (Abcam, Cat# ab115992, 3.5 mg/ml) diluted to 1:200 (in casein) for 1 h at room temperature. Slides were washed with Tris-buffered saline (TBS; containing 0.05% Tween-20, 0.01 M, PH 7.4) for 10 min and then incubated with 1.5% hydrogen peroxide solution (Fisher, Cat# H325-500) diluted in TBS-T for 5 min to dissolve all endogenous peroxide and then incubated with two drops of HRP-conjugated anti-rabbit secondary antibody (GBI Labs, Cat# D13-110) for 20 min at room temperature. Slides were washed in TBS-T for 10 min and incubated with an Alexa Fluor™ 488-conjuagted-tyramide signal amplification (TSA™, Invitrogen, Catalog# B40953) solution diluted to 1:500 in TBS-T for 10 min at room temperature to detect staining. Slides were blocked with two drops of RNAScope® Miultiplex FL v2 HRP blocker (ACD Life Sciences, Catalog# 323120) and incubated at 40 °C using a HybEZ™ hybridization oven (ACD Life Sciences) for 15 min. Slides were treated with an anti-HBs surface mouse monoclonal antibody (Clone A10F1, Biolegend, Cat# 932302, 0.5 mg/ml) diluted to 1:250 (in casein) for 1 h at room temperature. Slides were washed and incubated with the 1.5% peroxide solution (Fisher, Cat# H325-500) for 5 min followed by incubation with the 2-step HRP-conjugated anti-mouse secondary antibodies (GBI Labs, Cat# D37-110) for 20 min each at room temperature. Slides were washed in TBS-T for 10 min and then incubated with an Alexa Fluor™ 594-conjuagted-tyramide signal amplification (TSA™, Invitrogen, Catalog# B40957) solution diluted to 1:500 in TBS-T for 10 min at room temperature to detect staining. Slides were then subjected to antigen retrieval by microwaving using CA buffer solution. Slides were cooled and co-incubated for 1 h at room temperature with a cocktail of two antibodies: anti-CD68 (Clone KP1; Biocare Medical, Inc., Cat# CM033B) and anti-CD163 (Clone 10D6; Novocastra/ Leica Microsystems Inc., Cat# MA5-11458) both diluted to 1:400 (in casein) for detection of Kupffer cells in the liver. Slides were then incubated with 1.5%

peroxide (Fisher, Cat# H325-500), washed in TBS-T for 10 min and incubated with 2-step HRP-conjugated mouse secondary antibodies (GBI Labs, Cat# D37-110) for 20 min each at room temperature. Slides were washed in TBS-T for 10 min and incubated with an Alexa Fluor™ 647-conjuagted-tyramide signal amplification (TSA™, Invitrogen, Catalog# B40958) for 10 min at room temperature for fluorescence detection. Slides were washed in TBS-T for 10 min and then stained with DAPI (5 mg/ml, Invitrogen, Catalog# D1306) diluted to 1:10000 in TBS-T for 5 min at room temperature. Slides were washed, mounted with SlowFade™ Gold anti-fade reagent (Invitrogen, Catalog# S36937) and cover-slipped. Images were acquired by scanning the entire tissue at Å~40 magnification using an Olympus VS120 Slide Scanner, pseudocolored and analyzed using CellSens™ Dimension Desktop v1.18 software (Olympus).

**HBcAg and HBV RNA detection.** We combined vRNA detection with HBcAg immunofluorescence where both signals were detected by Tyramide Signal Amplification (TSA™, Invitrogen). First, liver samples were fixed, processed for paraffin embedding, slide sectioning and stained for HBcAg detection as described above. RNA detection was performed by using one custom probe set- one anti-sense (targeting vRNA transcripts)—covering different regions of the viral genome using the RNAscope® 2.0 HD Multiplex detection protocol (ACD Life Sciences) followed by Tyramide Signal Amplification (TSA™, Invitrogen) according to manufacturer's protocol. After HBcAg detection with TSA, slides were blocked with two drops of the RNAScope® Miultiplex FL v2 HRP blocker (ACD Life Sciences, Catalog# 323120) and incubated at 40 °C using a HybEZ™ hybridization oven (ACD Life Sciences) for 15 min followed by treatment with RNAScope® Protease III (ACD Life Sciences, Catalog# 322337), diluted 1:10 in PBS for 5 min at 40 C. Slides were rinsed twice with water and then incubated with 1.5% peroxide (Fisher, Cat# H325-500) for 5 min at room temperature. Slides were again rinsed with water and then incubated with a drop of pre-warmed customized HBV probe (V-HBV-AD38-P-01, ACD Life Sciences, Catalog# 513421) and incubated for 2 h at 40 °C. Slides were washed in wash buffer (0.1X SSC, 0.03% lithium dodecyl sulfate) and incubated with amplification reagents as described in the RNAScope® Miultiplex Fluorescence Detection Reagents v2 kit (ACD Life Sciences, Catalog# 323110). Slides were treated with Amplifier 1 (2 nmol/L) in hybridization buffer B (20% formamide, 5X SSC, 0.3% lithiumdodecyl sulfate, 10% dextran sulfate, blocking reagents) at 40 °C for 30 min; Amplifier 2 (a proprietary enhancer to boost detection efficiency) at 40 °C for 15 min; Amplifier 3 (2 nmol/L) in hybridization buffer B at 40 °C for 30 min; Amplifier 4 (2 nmol/L) in hybridization buffer C (2X SSC, blocking reagents) at 40 °C for 15 min; Amplifier 5 (a proprietary signal amplifier) at room temperature for 30 min; Amplifier 6 (a proprietary secondary signal amplifier) at room temperature for 15 min. After each hybridization step, slides were washed with wash buffer three times at room temperature. Before detection, the slides were rinsed one time in 1X TBS Tween-20 (0.05% v/v). Amplification 6 contained alkaline phosphatase (or horseradish peroxidase) labels, which was detected with an Alexa Fluor™ 594-conjuagted-tyramide signal amplification (TSA™, Invitrogen, Catalog# B40957) solution for 10 min at room temperature. Slides were washed in TBS-T for 10 min and then stained with DAPI (5 mg/ml, Invitrogen, Catalog# D1306) diluted to 1:10000 in TBS-T for 5 min at room temperature. Slides were washed, mounted with SlowFade™ Gold anti-fade reagent (Invitrogen, Catalog# S36937) and cover-slipped. Images were acquired by scanning the entire tissue at Å~40 magnification using an Olympus VS120 Slide Scanner and pseudocolored using the CellSens™ Dimension Desktop v1.18 software (Olympus).

**Immunofluorescence for detection of CD3 in livers.** Liver samples were fixed in 4% paraformaldehyde, processed for paraffin embedding, and cut at 7 μm thickness. The slides were de-waxed in xylene, re-hydrated in graded ethanol and then subjected to antigen retrieval in 1X citrate buffer pH 6 (GBI Labs, Catalog# B05C-100B) for 15 min at 110 °C and 5 psi. Slides were cooled, blocked in 0.25% Casein (Sigma Aldrich, Catalog# C7078) solution (in PBS) for 10 min at room temperature and incubated with a 1:400 dilution of an anti-CD3 rabbit antibody clone SP7 (Epredia Lab Vision #RM9107S) for 1 hr at room temperature. Slides were washed in TBS-T for 5 min and incubated with 1:250 dilution of a goat anti-rabbit IgG (H + L) antibody (Invitrogen #A21244) for 1 h at room temperature. Slides were washed in TBS-T for 5 min and then stained with DAPI (Invitrogen #D1306) diluted to 1:10,000 in TBS-T for 5 min at room temperature. Slides were washed, mounted with SlowFade™ Gold anti-fade reagent (Invitrogen, Catalog# S36937) and cover-slipped. Images were acquired by scanning the entire tissue at Å~40 magnification using an Olympus VS120 Slide Scanner, pseudocolored, and analyzed using the CellSens™ Dimension Desktop v1.18 software (Olympus).

**HBV Sequencing.** We performed a modification of a previously published HBV sequencing protocol. HBV DNA was isolated from liver and serum samples as described above. Concentrations of HBV DNA samples extracted from livers were normalized to 10 ng/μL via dilution in water prior to use. The concentrations of HBV DNA samples extracted from serum were not adjusted prior to use. To prepare isolated HBV DNA samples for deep sequencing, first the relaxed circular DNA (rcDNA) of the HBV genome was completed to a fully double-stranded circular genome. A mixture of 0.5 μL T4 DNA Polymerase (3 U/μL, New England Biolabs, Catalog# M0203S), 0.2 μL T4 DNA Ligase (1,000 U/μL, ThermoFisher

Scientific, Catalog# FEREL0011), 0.5 μL BSA (2 mg/mL, New England Biolabs, Catalog# B9000S), 0.5 μL dNTP Mix (10 mM each, New England Biolabs, Catalog# N0447S), and 0.1 μL ATP (100 mM, ThermoFisher Scientific, Catalog# FERR1441) in 0.7 μL water and 1 μL 10x phi29 DNA polymerase buffer (New England Biolabs, Catalog# B0269S) was added to each 6.5 μL DNA sample. The mixtures were then placed in a thermocycler at 30 °C for 45 min, then 75 °C for 20 min, and finally cooled to 10 °C. After the reaction, the fully dsDNA HBV genome underwent rolling circle amplification. 0.5 μL of an HBV-specific primer pool (12.5 μM each, Table S3) was added to each DNA template and placed in the thermocycler with the following protocol: 95 °C for 2 min, 50 °C for 15 sec, 30 °C for 15 sec, 20 °C for 10 min, and finally cooled to 4 °C. Next, the template was amplified into linear concatemers via the addition of the following mixture to each sample: 0.6 μL primer pool (12.5 μM each, Table S3), 0.4 μL dithiothreitol (100 mM, ThermoFisher Scientific, Catalog# 707265 ML), 1.5 μL BSA (2 mg/mL, New England Biolabs, Catalog# B9000S), 3.5 μL dNTP Mix (10 mM each, New England Biolabs, Catalog# N0447S), 0.1 μL pyrophosphatase (2 U/μL, New England Biolabs, Catalog# M0296S), and 1 μL phi29 DNA Polymerase (10 U/μL, New England Biolabs, Catalog# M0296L) in 1.3 μL water and 1 μL 10x phi29 DNA polymerase buffer (New England Biolabs, Catalog# B0269S). The reaction mixtures were placed in the thermocycler for the rolling circle amplification protocol: 30 °C for 22 h, then 65 °C for 15 min, with a final hold temperature of 10 °C. The HBV dsDNA concatemers were then isolated from the RCA mixture via the addition of 80 μL AMPure XP DNA Purification Beads (Beckman Coulter, Catalog# A63881) to each sample. The bead slurries were placed on a magnetic rack and the supernatants discarded. The beads were then washed two times with 200 μL 80% ethanol, then allowed to dry for ten minutes. The dried bead mixtures were thoroughly resuspended with 20 μL water then returned to the magnet. The clear supernatants were then transferred to new tubes and quantified using the Qubit dsDNA Broad Range Kit (ThermoFisher Scientific, Catalog# Q32850) on a Qubit 4.0 Spectrophotometer. The purified concatemers were normalized to 5 ng/μL using water, samples with concentrations under 5 ng/μL were not adjusted. The adjusted concatemers were then processed through the Twist Bioscience Protocol for library preparation per the manufacturer's instructions. Following Twist Bioscience library preparation, quantified serum-derived samples and liver-derived samples were pooled into separate DNA Lo-Bind tubes (Eppendorf, Catalog# 022431021) at equal nanogram values. The pooled samples were dried at 58 °C for 18 h. After drying, the two pooled samples were prepared for HBV probe hybridization with the addition of 5 μL Blocker Solution (Twist Bioscience, Catalog# PN100864) and 7 μL Universal Blockers (Twist Bioscience, Catalog# PN100865). Each pool was prepared by mixing 20 μL of Hybridization Mix (Twist Bioscience, Catalog# PN100587) and 4 μL of the Twist Custom HBV Panel (Twist Bioscience, Catalog# PN10100) and heating in the thermocycler at 95 °C (lid at 105 °C) for 2 min and then immediately cooled on ice for 5 min, then equilibrated to 25 °C. Next, the pools were heated in the thermocycler at 95 °C (lid at 105 °C) for 2 min, then equilibrated to 25 °C. Once both pools and probe solutions equilibrated, 28 μL of probe solution was added to each pool and vortexed. 30 μL of Hybridization Enhancer (Twist Bioscience, Catalog# PN100937) was layered over each probe capture reaction and the samples were incubated in the thermocycler at 70 °C (lid at 85 °C) for 18 h. HBV probe-bound DNA was then isolated using the manufacturer's instructions (Twist Biosciences). DNA was quantified using the Bioanalyzer High Sensitivity DNA Analysis Kit (Agilent, Catalog# 5067-4626) on an Agilent 2100 Bioanalyzer. After quantification, the two pools were combined into a single library at 2 nM. The sample was run on an Illumina Miseq using 2 × 250 paired-end reads. Illumina sequence data were processed using Geneious Prime software v2022.0.2 ensuring a sequencing depth of ≥ 50 reads per base and a Q-score ≥ 30.

**Statistics & Reproducibility.** No statistical methods were used to predetermine sample sizes. No data were excluded from the analyses. The experiments were not randomized and the Investigators were not blinded to allocation during experiments and outcome assessment.

**Reporting summary.** Further information on research design is available in the Nature Research Reporting Summary linked to this article.

## Data availability

Reference sequences used in this study can be accessed through GenBank accession numbers NM_003049.4 and U95551.1. Source data are provided with this paper.

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

## Acknowledgements

We would like to thank Skip Virgin, Christy Hebner, Florian Lempp, Andrea Cathcart, and the Vir Biotechnology Virology section for productive discussion pertaining to chronic HBV infection and model development. We would also like to thank Jake Estes and his lab for support in optimizing liver microscopy, and to thank all animal care and veterinary staff at the Oregon National Primate Research Center for their dedication and tireless efforts to ensure the ethical completion of our study. Funding was provided by Vir SRA-17-077 (BJB, JBS), the Burroughs Wellcome Fund Award 101539 (AP), and the National Institutes of Health R21 AI108401 (BJB), R01 AI144008 (BJB), R01 AI157612 (BJB), Health R01 AI129703 (JBS), R01 AI138797 (AP), R01 AI153236 (AP), P51 OD011092 (ONPRC).

## Author contributions

Conceptualization: J.B.S., B.J.B. Methodology: S.B., L.N.R., J.M.W., S.Y., B.J.B. Validation: B.J.B. Formal analysis: T.T., M.R.K., S.S., B.N.B., B.J.B. Investigation: S.B., L.N.R., J.M.W., S.Y., M.F., J.N.C., J.J., L.W., T.T., M.R.K., S.S., T.T., J.V.S., B.J.B. Resources: M.A., B.N.B., U.P., A.P., G.G. Writing - Original Draft: J.B.S, B.J.B Writing - Review & Editing: S.B., L.N.R., J.M.W., S.Y., U.P., A.P., G.G. Visualization: B.J.B. Supervision: S.S., U.P., A.P., J.V.S., G.G., B.J.B. Project administration: B.J.B. Funding acquisition: A.P., J.B.S., B.J.B.

## Competing interests

Authors have no competing interests.
