## [Peer Review File · Nature Communications]

REVIEWER COMMENTS

Reviewer #1 (Remarks to the Author):

In this report, Biswas et al. further characterize the use of rhesus macaques (RM) as a model of transient and prolonged HBV infection. The findings are interesting and the establishment of a model of prolonged HBV infection, albeit cumbersome and not completely defined, could have a significant impact for the scientific community and the developers of novel therapeutic strategies against chronic hepatitis B in humans. Some issues remain and they are listed below:

General comment

It would be much better to soften the statements that a model of chronic HBV infection has been established. Even though 3 out of 3 heavily immunosuppressed animals showed signs of active HBV replication for over 6 months post infection, only one of them remained weakly viremic past week 40. Further, all 3 animals seroconverted within that time frame and this is something that doesn't happen in chronically infected patients. Using terms such as prolonged HBV infection (rather than chronic HBV infection) would be preferable.

Fig. 1

Fig.1C. It would be nice to compare the relative amount of hNTCP RNA copies/100 ng in the RM livers with similar data obtained in hNTCP-expressing cell lines (just to have a rough comparison on the absolute levels of hNTCP expression one can get after the in vivo Ad5-based transduction)

2. Any presence of hepatocytes still positive for HBcAg in the liver biopsies obtained either at 6 wpi or at euthanasia?

Fig. 2.

Fig. 2A Given the potential presence of cross-reactive antibodies or potential issues with non-specific binding in the assay, in addition to show seroconversion it would be informative to also depict HBsAg loss overtime

Fig. 2B The authors should clearly indicate which are the individuals showing whatever T cells response, linking that individual response to the individual sALT response and the individual profiling showed below in panels C-H.

Fig. 2C-H. Looks like the expression analysis of immunomodulatory genes in these panels is limited to 3 animals per time point. Are these the 3 juveniles showing elevated serum ALT? If so, what happened to the others?

Fig. 3.

The readership would appreciate a sentence (possibly in the Discussion) commenting how/why this more stringent immunosuppression regimen has worked better, taking advantage of past experience with the allogeneic stem cell transplant models. Along these lines, It would have been instructive to study here blood- and liver-derived immune parameters similar to those depicted in Fig. 3B-H (particularly at time points shortly before and after tapering). Not clear why this was not done.

Fig. 4.

Here again it would have been nice to look for some immune-related parameters of adaptive immunity (e.g., CD3, CD4 and CD8) by IF/IHC in the longitudinal liver biopsies.

Fig. 4H: Could the authors include/substitute a graph calculating the % of HBcAg+ hepatocytes over the % of total hepatocytes? This would read much better than the % of HBcAg+ hepatocytes per square mm.

Minor:

1. The authors should change the (confusing/overlapping) coloring and dashing of lines depicting results from individual RM in Figs 1B,E and Figs 2A.
2. Please avoid to describe with “high-level viremia” titers that are often way below 10^8 genomes/ml. High levels of viremia in HBV infection generally refer to titers above that threshold.

Reviewer #2 (Remarks to the Author):

This manuscript described their experiments using Rhesus monkey (RM) to develop persistent HBV infection. Basically, the authors repeated their previous paper (published in Nature COmmunication 2017) by transiently introducing hNTCP into the RM liver before HBV infection. Previously, they barely observed low HBV infections in the liver or in the blood. Now they tried to increase HBV infection efficiency by using juvenile RM, or pre-conditioning them with immuno-suppressive agents. Despite of these changes, all RM cleared HBV after 6 months (no matter in juvenile or in immuno-suppressed RMs). Therefore, the results were similar to previous publication and nothing new.

Major points:

1. The HBV research community looked forward to this group about the hNTCP transgenic RM as a promising model for HBV persistence. Their previous transient hNTCP expression in RM suggested this possibility. Current study failed to deliver solid and reliable RM model for chronic HBV.
2. The use of several immuno-suppressing agents, such as dexamethasone, antiCD20 or Tacrolimus, to increase HBV persistence, is only relevant to the HBV infection in immuno-compromised hosts (such as organ transplantation recipients or hemodialysis patients). Even such pre-conditioning regimens help the establishment of persistent HBV in RM, the involved immune mechanisms and pathogenesis different from most HBV persistence that occur in newborns without immuno-deficiency.

Specific points.

1. The experiments data failed to justify the title: chronic hepatitis B infection---. As the HBV infection in juvenile or immuno-suppressed RM failed to maintain HBV infection more than 6 months.
2. As the number of studied RM in each experiment was so low, the data varied a lot and difficult to reproduce.
3. The assay for innate immune response in the liver of RM showed different expectations, as it is a well-known fact that HBV infection triggered little conventional innate immune responses. Can the authors distinct the elevated expression of ISG, TNFalpha or CXCL10, response induced by Adenovirus/AAV or by HBV ?
4. The expression of hNTCP and HBcAg in the RM liver (in last figure) was poorly presented and difficult to be convincing. Most importantly the co-localization was essential.
5. The assay for cccDNA in the last figure and in the supplement was not described in details.

Reviewer #3 (Remarks to the Author):

A major limitation in HBV research is the lack of immune competent models permissive for HBV infection and developing chronic hepatitis. In an attempt to generate experimental in vivo models fulfilling these criteria, the authors previously demonstrated that exogenous expression of hNTCP in Rhesus macaque (RM) enable HBV infection, although HBV replication was very short-lived, without

clear detection of serological markers such as HBsAg and HBeAg, and rapid resolution. To increase infection efficiency and aiming to achieve chronicity, in this study not only juveniles and infant animals were injected with Ad5 or AAV8 expressing the human NTCP but also immunosuppressive tacrolimus regimen, liposomal alendronate to deplete Kupffer cells and dexamethasone injections were used to lower immune responses before injecting high titers HBV (1×10^9 copies). This highly suppressive strategy led to transient development of viremia (up to 10^6 HBV DNA copies/ml). Analysis of immunity markers confirmed infection clearance in all animals and anti-HBs seroconversion in 2/5 macaques. Maintenance of immune suppression in the weeks following HBV infection (till day 126), prolonged HBV replication markers in all 3 animals. Upon withdrawal of immunosuppressants, 2/3 animals efficiently cleared HBV, while 1 animal appeared to maintain some HBV markers (HBeAg) despite HBsAg negativization. Clearance kinetics correlated with loss of hNTCP expression.

In general, the manuscript is well written and it clearly shows how continuous administration of strong immunosuppressants can improve both infection levels and duration in MR transduced with viral vectors expressing hNTCP. A major limitation of the study design is that a very low number of animals was used, with animals that were even used for HBV/AAV re-challenge attempts ($n=2$). Moreover, while knowledge of the approach used and results obtained is important for the scientific community, the utility of the RM for preclinical immune pathogenesis studies remains unclear.

Specific comments

1) Fig.1 and Fig.S3: It seems that NTCP expression levels correlate with HBV intrahepatic loads, suggesting that the amount of NTCP positive cells determines infection levels achievable. Moreover, infection plateau are achieved surprisingly fast (within 4 weeks), suggesting that intrahepatic spreading in the first weeks post infection may not play a major role. At present, this is unclear and it would be interesting to gain some knowledge regarding spreading kinetics in the first weeks post infection. Could the authors provide some intrahepatic staining (HBcAg distribution among NTCP positive cells) from liver samples obtained at earlier time points?

2) It also remains unclear whether infection loads may affect the strength of the immune responses determined later (i.e. HBs-seroconversion). It would be interesting if the authors would provide this type of information. Unfortunately it is difficult to appreciate such correlations from the small figures 1 and 2 provided and the colour code used.

3) Figure 2C-H. Did the so-called naïve controls receive the AAV/AD5 injections and identical suppressive treatment? Please clarify since such animals would serve as proper controls.

4) Fig. 3 Continuous administration of immune suppressive regimens maintained HBV replication markers for 6 months. However, these markers - in particular HBsAg and HBeAg - dropped fast after

stopping treatment with only 1/3 animals showing a slower decrease. Since the other 2 animals were used for re-challenge experiments, the different (additional) experimental design should be presented clearly (i.e. in Fig.3A) or data obtained in individual animals shown separately.

5) As mentioned above, the authors should enlarge figures 1-3.

6) Fig.4 G. it is impossible to appreciate the RNA scope IHC staining. Please provide higher magnification (close up?) of both figure 4F and G.

7) The authors also claim “We also included staining for macrophages (CD68/CD163) and confirmed that Kupffer cells showed no signs of HBV replication”. This is however impossible to see from the figures provided. Again, add panels with higher magnification pointing this out.

8) HBV re-challenge experiments failed indicating occurrence of acquired anti-HBV immunity. However, since reinfection depends also on the efficiency of AAV-hNTCP infection establishment, it would be important to know if an immune response was also developed against the viral vector. Please comment.

9) Moreover, do the authors have evidence that hNTCP pos. cells were present at the time of HBV reinfection? As mentioned above, visualization of hNTCP positive hepatocytes would be helpful. If human specific staining is not possible because of cross-reactions with macaque NTCP, the authors could provide these data by using an RNA-scope approach (see also fig.1 comments).

10) Discussion: “... given the relatively small size of our juvenile RM we utilized all available tissues to characterize the markers of HBV infection.” I would delete this sentence, since knowing how many different analyses can be done even with small animals like mice, the argument does not sound convincing.

11) As recognised by the authors, the main barrier of this model is the prompt immune clearance of HBV upon withdrawal of immunosuppression. Thus, key immunological studies may remain limited in a system lacking HBV-induced exhaustion of immune responses. Nevertheless, to emphasize the model, the authors wrote that they “believe this new RM model will be transformative”. I would strongly encourage the authors to tune down this type of statements and to be more critical considering the strong limitations the model currently has.

12) „.. through downregulation of RIG-I, SMC5/6, and other pattern recognition receptors”. Add downregulation and/or degradation of ... to be more precise.

RESPONSE TO REFEREES

Reviewer #1 (Remarks to the Author):

In this report, Biswas et al. further characterize the use of rhesus macaques (RM) as a model of transient and prolonged HBV infection. The findings are interesting and the establishment of a model of prolonged HBV infection, albeit cumbersome and not completely defined, could have a significant impact for the scientific community and the developers of novel therapeutic strategies against chronic hepatitis B in humans. Some issues remain and they are listed below.

We thank the reviewer for the positive feedback and we fully agree that while the work described herein is a critical step in the generation of an NHP model of HBV infection, improvements must continue to be made. While outside the immediate scope of this manuscript, our group is currently undertaking studies to more thoroughly identify the immune responses associated with HBV clearance in rhesus macaques, as driving tolerance to HBV in fully immunocompetent rhesus macaques is our overarching goal.

General comment

It would be much better to soften the statements that a model of chronic HBV infection has been established. Even though 3 out of 3 heavily immunosuppressed animals showed signs of active HBV replication for over 6 months post infection, only one of them remained weakly viremic past week 40. Further, all 3 animals seroconverted within that time frame and this is something that doesn't happen in chronically infected patients. Using terms such as prolonged HBV infection (rather than chronic HBV infection) would be preferable.

Thank you for this comment. We agree that there are differences between the rhesus macaque model and human HBV infection. Our original intention was to use the clinical definition of chronic HBV infection (HBsAg or HBV DNA in blood or liver for greater than 6 months), which we do attain, but the reviewer's comment is correct in that differences do exist in immunity to and serology of HBV infection between the species. We have dampened our language to more appropriately categorize the infection we see in rhesus macaques, replacing the adjective "chronic" with adjectives such as "prolonged", "long-term", "sustained", and "extended".

Fig. 1

Fig.1C. It would be nice to compare the relative amount of hNTCP RNA copies/100 ng in the RM livers with similar data obtained in hNTCP-expressing cell lines (just to have a rough comparison on the absolute levels of hNTCP expression one can get after the in vivo Ad5-based transduction)

We now include information in the text indicating the quantification of hNTCP RNA transcripts found in an equal mass of RNA taken from a HepG2-hNTCP cell line. Lines 111-114.

2. Any presence of hepatocytes still positive for HBcAg in the liver biopsies obtained either at 6 wpi or at euthanasia?

Understanding transient HBV infection in rhesus macaques is a primary goal of our group, both because it will open doors to better chronic (or extended) HBV infection models but also because it may lead to important information on adult human clearance of HBV in the vast majority of cases. Therefore, we thank the reviewer for this comment, since we did not originally include this information on HBV antigen expression in transiently viremic animals. We now include Fig. S2 indicating that RM 36651, the animal with the highest peak sVL, had HBcAg+ nuclei in the liver at 6wpi, but not at necropsy following clearance. Lines 108-110.

Fig. 2.

Fig. 2A Given the potential presence of cross-reactive antibodies or potential issues with non-specific binding in the assay, in addition to show seroconversion it would be informative to also depict HBsAg loss overtime.

We apologize for the oversight of not discussing HBsAg levels in rhesus macaques with transient viremia. We have found that HBsAg is only detectable in HBV-infected rhesus macaques when viral loads exceed $\sim 1 \times 10^4$ copies/mL. We now include Fig. S1 showing that HBsAg was detected in three RM that exceeded this threshold. Lines 105-106.

Fig. 2B The authors should clearly indicate which are the individuals showing whatever T cells response, linking that individual response to the individual sALT response and the individual profiling showed below in panels C-H.

We thank the reviewer for pointing out that the T cell response information shown in Fig. 2B,C was disorganized. We now re-organized the T cell response data by animal, so that correlations with sALT can be assessed.

Fig. 2C-H. Looks like the expression analysis of immunomodulatory genes in these panels is limited to 3 animals per time point. Are these the 3 juveniles showing elevated serum ALT? If so, what happened to the others?

We thank the reviewer for the comment, and we note that both reviewers 2 and 3 also pointed out that this experiment looking into liver immunity did not include a control group receiving Ad only. Because HBV is known to be a stealth virus in humans, and given the lack of appropriate controls, we have made the decision to remove this figure panel from the manuscript. The conclusion that HBV is not a stealth virus in rhesus macaques cannot be substantiated with the current data, given the confounding variable of high-titer, intravenous Ad injection.

Fig. 3.

The readership would appreciate a sentence (possibly in the Discussion) commenting how/why this more stringent immunosuppression regimen has worked better, taking advantage of past experience with the allogeneic stem cell transplant models. Along these lines, it would have been instructive to study here blood- and liver-derived immune parameters similar to those depicted in Fig. 3B-H (particularly at time points shortly before and after tapering). Not clear why this was not done.

We thank the reviewer for these comments. This stringent immunosuppression regimen successfully suppresses adaptive immunity in allogeneic stem cell transplants, even if donor and recipient are MHC-mismatched. Thus, it is one of the strongest immunosuppression regimens that has been previously tested in NHPs (and one of the strongest in humans as well). The idea behind using such immunosuppression was a “work backwards” approach, whereby we showed that immunity is key to elimination of HBV in rhesus macaques and we are now interrogating specific immune subsets. However, we believe this work is outside the scope of the present paper. We now discuss this immunosuppression regimen in more detail in the discussion, with references to our experience in the stem cell transplantation field. Lines 289-292.

We also have significantly expanded our immunological analysis of RM before and after immunosuppression. This data includes additional data looking at longitudinal anti-HBV T cell responses in the blood of animals with extended viremia in Fig. 3H-J. This HBV-specific T cell profile is accompanied by Ki67 phenotyping of both CD4+ and CD8+ T cells in blood before and after immunosuppression in Fig. 3K. Lines 169-176.

Fig. 4.

Here again it would have been nice to look for some immune-related parameters of adaptive immunity (e.g., CD3, CD4 and CD8) by IF/IHC in the longitudinal liver biopsies.

We now include longitudinal IFA staining of liver biopsies for CD3+ T cells in Fig. 4I. We found an increase in CD3+ T cells in the liver post-immunosuppression, concomitant with the increase in Ki67+CD3+ T cells in the blood. Lines 214-219.

Fig. 4H: Could the authors include/substitute a graph calculating the % of HBcAg+ hepatocytes over the % of total hepatocytes? This would read much better than the % of HBcAg+ hepatocytes per square mm.

We thank the reviewer for the comment. The vast majority of cells in our liver staining are hepatocytes (>90%). We therefore now include this new analysis in Fig. 4H, indicating % of total cells that are HBcAg+. Lines 207-213.

Minor:

1. The authors should change the (confusing/overlapping) coloring and dashing of lines depicting results from individual RM in Figs 1B,E and Figs 2A.

We thank the reviewers for pointing out the confusing color and graph schema and we have adjusted all of the figures accordingly.

2. Please avoid to describe with “high-level viremia” titers that are often way below 10^8 genomes/ml. High levels of viremia in HBV infection generally refer to titers above that threshold.

We have updated the text to remove “high-level viremia”.

Reviewer #2 (Remarks to the Author):

This manuscript described their experiments using Rhesus monkey (RM) to develop persistent HBV infection. Basically, the authors repeated their previous paper (published in Nature Communication 2017) by transiently introducing hNTCP into the RM liver before HBV infection. Previously, they barely observed low HBV infections in the liver or in the blood. Now they tried to increase HBV infection efficiency by using juvenile RM, or pre-conditioning them with immuno-suppressive agents. Despite of these changes, all RM cleared HBV after 6 months (no matter in juvenile or in immuno-suppressed RMs). Therefore, the results were similar to previous publication and nothing new.

We respectfully disagree with this reviewer that the results are “similar to [our] previous publication and nothing new”. To our knowledge, no-one has previously demonstrated that robust HBV (not surrogate viruses) viremia can be attained in small NHP species. While it was necessary to immunosuppress the animals to establish long-term infections, this pre-clinical model can be immediately used to test direct targeting anti-HBV drugs (capsid inhibitors, cccDNA CRISPR/Cas, etc.). In addition, the presented study lays the critical groundwork showing that it is adaptive immunity that clears HBV infection in rhesus macaques, similar to HBV clearance in humans. Therefore, we believe the manuscript is highly relevant to the HBV field.

Major points:

1. The HBV research community looked forward to this group about the hNTCP transgenic RM as a promising model for HBV persistence. Their previous transient hNTCP expression in RM suggested this possibility. Current study failed to deliver solid and reliable RM model for chronic HBV.

We understand that there are great hopes surrounding the creation of transgenic macaques wherein 100% of hepatocytes express the hNTCP receptor. However, the creation of transgenic non-human primates is a slow and arduous process, and breeding of these transgenic monkeys will require an additional 5 years given the age of sexual maturity in rhesus macaques. Thus, having an NHP model of HBV infection “in-hand”, as we describe here, is still a major advancement. While immunosuppression does preclude the testing of some therapeutics, the model can be used readily for the study of direct acting antivirals such as CRISPR/Cas9 targeting cccDNA, capsid inhibitors, etc., and we now make this point more openly in the Discussion. Lines 321-322. Thus, we believe there is still significant merit to our model.

With this being said, we recently welcomed our first hNTCP transgenic rhesus macaque, a male named Tauro, at ONPRC in August of 2021. This animal will reach sexual maturity in about 3 years. Description of this animal, however, is outside the scope of this manuscript.

It should also be noted that there is no guarantee that hNTCP transgenic macaques will progress to chronic HBV infection following challenge, and building forward from this study to understand what components of the immune system inhibit HBV replication in rhesus macaques will inform future iterations of the model, regardless if that model is hNTCP transgenic or not.

2. The use of several immuno-suppressing agents, such as dexamethasone, anti-CD20 or Tacrolimus, to increase HBV persistence, is only relevant to the HBV infection in immuno-compromised hosts (such as organ transplantation recipients or hemodialysis patients). Even such pre-conditioning regimens help the establishment of persistent HBV in RM, the involved immune mechanisms and pathogenesis different from most HBV persistence that occur in newborns without immuno-deficiency.

We agree that it remains unclear how similar NHP and human HBV infection is in regards to immunity and pathogenesis, but would argue that any new model must be defined over multiple iterations. There are several reasons immunity and pathogenicity may differ, including the use of viral vectors for hNTCP expression or the fact that challenged neonates are not born to HBV+ mothers (and therefore are not exposed to HBV antigen in utero). Overall, we are convinced that the weaknesses of the model are outweighed by the fact that this remains the only robust NHP model of HBV replication currently available.

Specific points.

1. The experiments data failed to justify the title: chronic hepatitis B infection---. As the HBV infection in juvenile or immuno-suppressed RM failed to maintain HBV infection more than 6 months.

We agree with reviewers 1 and 2 and have adjusted not only the title, but also the text, to indicate that there are differences to chronic infection as seen in humans. However, we believe that any animal model has differences to human infection. Please see our response to reviewer 1.

2. As the number of studied RM in each experiment was so low, the data varied a lot and difficult to reproduce.

The concept that large group sizes are required to show important and relevant data was recently discussed by Bacchetti et al. in Science Translational Medicine in a paper entitled, "Breaking Free of Sample Size Dogma to Perform Innovative Translational Research". In this opinion piece, the authors clearly define why small sample sizes, particularly during early-stage research such as model development, is actually an acceptable and oftentimes preferable option. We trust that this reviewer agrees that our study fits perfectly into this definition. Non-human primate research is always offset by both the cost and ethics of NHP usage, and we believe that the critical finding -- RMs replicate HBV to clinically relevant levels and for extended periods of time under immunosuppression -- is a major advancement despite the small number of RMs used. We agree with the reviewer that mention of the animal numbers used is an important discussion point and text has been added. **Lines 269-270.**

3. The assay for innate immune response in the liver of RM showed different expectations, as it is a well-known fact that HBV infection triggered little conventional innate immune responses. Can the authors distinct the elevated expression of ISG, TNFalpha or CXCL10, response induced by Adenovirus/AAV or by HBV ?

Please see our response to reviewer 1, who also pointed out that proper "Ad only" controls were absent from Fig. 2 innate immune qPCR data. Given the confounding variable of Ad injection, we have removed the figure panels from the manuscript.

4. The expression of hNTCP and HBcAg in the RM liver (in last figure) was poorly presented and difficult to be convincing. Most importantly the co-localization was essential.

We thank the reviewer for the comment. During the submission process we lost significant resolution in our figures, which impacted the microscopy especially. We have now resolved the issue and provide high-resolution images of markers of HBV infection in the liver. In regards to co-localization, we are currently testing a next-generation adenovirus whereby hNTCP is codon-optimized and FLAG-tagged, allowing for colocalization of HBV antigens and hNTCP. However, given the wealth of knowledge showing that RM hepatocytes are not susceptible to HBV infection and that hNTCP expression is required for HBV infection, we believe that it can be safely inferred that hNTCP expression is the driver of successful HBV infection in our model.

5. The assay for cccDNA in the last figure and in the supplement was not described in details.

The entire protocol is outlined in the methods on Lines 466-525.

Reviewer #3 (Remarks to the Author):

A major limitation in HBV research is the lack of immune competent models permissive for HBV infection and developing chronic hepatitis. In an attempt to generate experimental in vivo models fulfilling these criteria, the authors previously demonstrated that exogenous expression of hNTCP in Rhesus macaque (RM) enable HBV infection, although HBV replication was very short-lived, without clear detection of serological markers such as HBsAg and HBeAg, and rapid resolution. To increase infection efficiency and aiming to achieve chronicity, in this study not only juveniles and infant animals were injected with Ad5 or AAV8 expressing the human NTCP but also immunosuppressive tacrolimus regimen, liposomal alendronate to deplete Kupffer cells and dexamethasone injections were used to lower immune responses before injecting high titers HBV (1x10⁹ copies). This highly suppressive strategy led to transient development of viremia (up to 10⁶ HBV DNA copies/ml). Analysis of immunity markers confirmed infection clearance in all animals and anti-HBs seroconversion in 2/5 macaques. Maintenance of immune suppression in the weeks following HBV infection (till day 126), prolonged HBV replication markers in all 3 animals. Upon withdrawal of immunosuppressants, 2/3 animals efficiently cleared HBV, while 1 animal appeared to maintain some HBV markers (HBeAg) despite HBsAg negativization. Clearance kinetics correlated with loss of hNTCP expression.

In general, the manuscript is well written and it clearly shows how continuous administration of strong immunosuppressants can improve both infection levels and duration in MR transduced with viral vectors expressing hNTCP. A major limitation of the study design is that a very low number of animals was used, with animals that were even used for HBV/AAV re-challenge attempts (n=2). Moreover, while knowledge of the approach used and results obtained is important for the scientific community, the utility of the RM for preclinical immune pathogenesis studies remains unclear.

We appreciate the reviewer's comments, which echo the comments from reviewer 2 concerning the number of animals utilized in the study. Please see our response to reviewer 2 regarding this topic. We are convinced that the main point of the paper – that RM replicate HBV long-term under immunosuppression and are a model immediately ready for testing of antivirals – is successfully made with the animal numbers provided.

Specific comments

1) Fig.1 and Fig.S3: It seems that NTCP expression levels correlate with HBV intrahepatic loads, suggesting that the amount of NTCP positive cells determines infection levels achievable. Moreover, infection plateau are achieved surprisingly fast (within 4 weeks), suggesting that intrahepatic spreading in the first weeks post infection may not play a major role. At present, this is unclear and it would be interesting to gain some knowledge regarding spreading kinetics in the first weeks post infection. Could the authors provide some intrahepatic staining (HBcAg distribution among NTCP positive cells) from liver samples obtained at earlier time points?

We thank the reviewer for the comment. We are also interested in early events in the RM liver post-HBV, since early events likely shape the overall outcome of infection in this model. We now include earlier HBcAg staining at 2 wpi in Fig. 4H. We detected no HBcAg+ cells at 2 wpi, despite the presence of HBV RNA in biopsies obtained from the same anatomical liver location. This was similar to 6 wpi,

where RNA was detected, but not HBcAg+ cells. In contrast, subsequent consecutive biopsies showed a large number of HBcAg+ cells, although there was inter-animal variability with RM 36901 still having no HBcAg+ cells at 10 wpi.

As mentioned in response to review 2, we do not currently have co-localization data as we are optimizing in vivo use of a next-generation adenovirus whereby hNTCP is codon-optimized and FLAG-tagged, allowing for colocalization of HBV antigens and hNTCP.

2) It also remains unclear whether infection loads may affect the strength of the immune responses determined later (i.e. HBs-seroconversion). It would be interesting if the authors would provide this type of information. Unfortunately it is difficult to appreciate such correlations from the small figures 1 and 2 provided and the colour code used.

*While we did not identify any correlation between the magnitudes of sVL and anti-HBs in the blood, we did see a trend between duration of viremia in the blood (time from first detectable HBV DNA to first time point below limit of quantification) and the magnitude of the anti-HBs response. We now show this data in Fig. S4B and discuss on **Lines 165-166**.*

We apologize for the poorly colored graphs, which also were presented at a lower resolution than hoped for. We have updated both the color scheme and the resolution of our figures, which hopefully will allow for easier assessment.

3) Figure 2C-H. Did the so-called naïve controls receive the AAV/AD5 injections and identical suppressive treatment? Please clarify since such animals would serve as proper controls.

All three reviewers commented on Figure 2C-H, which showed qPCR of IFN-stimulated genes, but did not include an Ad-only control group. Please see our response to reviewer 1. We have removed this data from Fig. 2.

4) Fig. 3 Continuous administration of immune suppressive regimens maintained HBV replication markers for 6 months. However, these markers - in particular HBsAg and HBeAg - dropped fast after stopping treatment with only 1/3 animals showing a slower decrease. Since the other 2 animals were used for re-challenge experiments, the different (additional) experimental design should be presented clearly (i.e. in Fig.3A) or data obtained in individual animals shown separately.

*We thank the reviewer for the comment and agree that Table S2 is large and makes timing of the additional procedures difficult to interpret. We now include the exact timing of these additional procedure in the main text, so that the reader can easily compare against the main figures. **Lines 177-187**.*

5) As mentioned above, the authors should enlarge figures 1-3.

We apologize for the small size of the figures, which were generated specifically to fit the column sizes of a final publication. We have expanded the size of the figures to make assessment easier.

6) Fig.4 G. it is impossible to appreciate the RNA scope IHC staining. Please provide higher magnification (close up?) of both figure 4F and G.

When converting to a PDF for initial review, we lost resolution on microscopy images and failed to recognize this issue. We have now rectified the resolution issue and hope this will allow for appropriate analysis.

7) The authors also claim “We also included staining for macrophages (CD68/CD163) and confirmed that Kupffer cells showed no signs of HBV replication”. This is however impossible to see from the figures provided. Again, add panels with higher magnification pointing this out.

Please see our response to comment 6. In addition to improving the image resolution, we have also increased the size of all figures. We believe it should now be clear that macrophages (yellow) do not stain for either HBsAg or HBcAg.

8) HBV re-challenge experiments failed indicating occurrence of acquired anti-HBV immunity. However, since reinfection depends also on the efficiency of AAV-hNTCP infection establishment, it would be important to know if an immune response was also developed against the viral vector. Please comment.

We have previously shown that animals treated with hNTCP vectors (AAV and HDAd) do not generate hNTCP specific responses (Burwitz et al, Nat Comm, 2017). In addition, we now include data in Fig. S4D showing that at the time of Ad-hNTCP administration these RMs did not express Ad5 neutralizing antibodies. Lines 181-182 and 185-186.

9) Moreover, do the authors have evidence that hNTCP pos. cells were present at the time of HBV reinfection?

Yes, as shown in Fig. 4A RM 37534 had detectable hNTCP expression at the time of HBV rechallenge (53 wpi), and this RNA level was similar to the levels seen during acute HBV infection in RM 36901.

10) Discussion: "... given the relatively small size of our juvenile RM we utilized all available tissues to characterize the markers of HBV infection." I would delete this sentence, since knowing how many different analyses can be done even with small animals like mice, the argument does not sound convincing.

We thank the reviewer for this comment and have adjusted the text accordingly.

11) As recognised by the authors, the main barrier of this model is the prompt immune clearance of HBV upon withdrawal of immunosuppression. Thus, key immunological studies may remain limited in a system lacking HBV-induced exhaustion of immune responses. Nevertheless, to emphasize the model, the authors wrote that they "believe this new RM model will be transformative". I would strongly encourage the authors to tune down this type of statements and to be more critical considering the strong limitations the model currently has.

We have adjusted the text according to the reviewer's comment. Line 317.

12) "... through downregulation of RIG-I, SMC5/6, and other pattern recognition receptors". Add downregulation and/or degradation of ... to be more precise.

We have adjusted the text according to the reviewer's comment. Line 277-279.

REVIEWERS' COMMENTS

Reviewer #1 (Remarks to the Author):

The revised manuscript has improved significantly. I'd recommend acceptance.

Reviewer #3 (Remarks to the Author):

The authors sufficiently addressed all questions and criticisms raised. Most importantly, they substantially extended the analyses and therefore greatly improved the characterization of the model. It is also now clearer what can be achieved and where the limitations remain both in terms of HBV infection, replication and responses that are possibly involved in viral clearance. Thus, the work is technically solid.

Main only remark is that despite all improvements, the model remains very complex and substantial variations were observed from animal to animal despite the strong immunosuppressive regimens. To cope with these variations, a larger number of animals need to be used for preclinical antiviral drug testing. This is why the authors may be more cautious about claiming in the discussion that the model is immediately ready for testing of antivirals. Perhaps, it could be honestly written that "further studies are needed to explore the potential of the model for testing antivirals".

RESPONSE TO REFEREES

Reviewer #1 (Remarks to the Author):

The revised manuscript has improved significantly. I'd recommend acceptance.

We appreciate the help from the reviewer improving the article.

Reviewer #3 (Remarks to the Author):

The authors sufficiently addressed all questions and criticisms raised. Most importantly, they substantially extended the analyses and therefore greatly improved the characterization of the model. It is also now clearer what can be achieved and where the limitations remain both in terms of HBV infection, replication and responses that are possibly involved in viral clearance. Thus, the work is technically solid.

Main only remark is that despite all improvements, the model remains very complex and substantial variations were observed from animal to animal despite the strong immunosuppressive regimens. To cope with these variations, a larger number of animals need to be used for preclinical antiviral drug testing. This is why the authors may be more cautious about claiming in the discussion that the model is immediately ready for testing of antivirals. Perhaps, it could be honestly written that “further studies are needed to explore the potential of the model for testing antivirals”.

We have updated the text in response to the reviewers comment. Line 323-324.